# UniCoTT: A Unified Framework for Structural Chain-of-Thought Distillation

**Xianwei Zhuang**[*], **Zhihong Zhu**[*], **Zhichang Wang**[*], **Xuxin Cheng, Yuexian Zou** [†]
School of Electronic and Computer Engineering, Peking University
`xwzhuang@stu.pku.edu.cn`

## Abstract

Chains of thought (CoTs) have achieved success in enhancing the reasoning capabilities of large language models (LLMs), while their effectiveness is predominantly observed in LLMs. Existing solutions methods adopt distillation to inject chain-of-thought capabilities into small models (SLMs). However, they: (1) can not guarantee the rationality of the generated explanation due to hallucinations; (2) ignore diverse structures of CoT during knowledge transfer. In this paper, we propose a unified CoT distillation framework termed UniCoTT for considering diverse structural CoTs (*i.e.*, chain, tree, and graph). UniCoTT contains two core strategies: iterative construction for structured CoTs and the structural constraint strategy. Specifically, UniCoTT prompts LLMs to iteratively produce accurate explanations with answers and unifies structured explanations as UniCoT which is seen as a bridge for knowledge transfer. Furthermore, UniCoTT utilizes the proposed unified supervised learning and structural consistency learning strategies to transfer knowledge of structured CoT to SLMs. Experimental results show that UniCoTT can significantly improve the performance of SLMs on multiple datasets across different NLP tasks. Our code is available at `https://github.com/mengchuang123/UniCoTT`.

## 1 Introduction

Large Language Models (LLMs) have demonstrated remarkable success across a wide range of textual tasks, requiring only a few examples as prompts (Brown et al., 2020; Nye et al., 2021), such as question-answer reasoning and natural language understanding (NLU). These models have been shown to address complex reasoning challenges by generating a step-by-step inference process, known as Chains of Thought (CoT)(Wei et al., 2022b;a). However, the enhancement of reasoning and question-answering capabilities through CoT prompts has been predominantly observed in LLMs (with over 100B parameters) (Wei et al., 2022b;a), which is not present in small language models (SLMs). Moreover, due to hallucinations (Ji et al., 2022), it is difficult to ensure that the generated reasons are consistent with the actual results (Maynez et al., 2020), nor can it guarantee the rationality of decisions (Wang et al., 2023), which affects the usability of the generated explanations.

The existing efforts (Maynez et al., 2020; Li et al., 2023) to distill the chain-of-thought capabilities from LLM into SLM[1] primarily involve learning from the outputs of LLMs. This process entails prompting LLMs (acting as teachers) to generate reasoning explanations for downstream datasets, which are subsequently used to train SLMs (acting as students) utilizing cross-entropy loss. However, these approaches overlook the fact that LLMs often produce text unrelated to the answers due to hallucinations (Ji et al., 2022). Consequently, the teacher models may not always generate reasoning explanations that comprehensively support the given answers (Wang et al., 2023).

In addition, current studies indicate that human reasoning processes involve complex mechanisms of backtracking and feedforward logic (Sloman, 1996; Yao et al., 2023) and develop intricate networks of thought, such as tree or graph reasoning paths (Dougherty & Franco-Watkins, 2000; Besta et al.,

---

[*]Equal contribution
[†]Corresponding author
[1]In this work, SLM mainly refers to the pre-training language model whose model size is less than 1B.

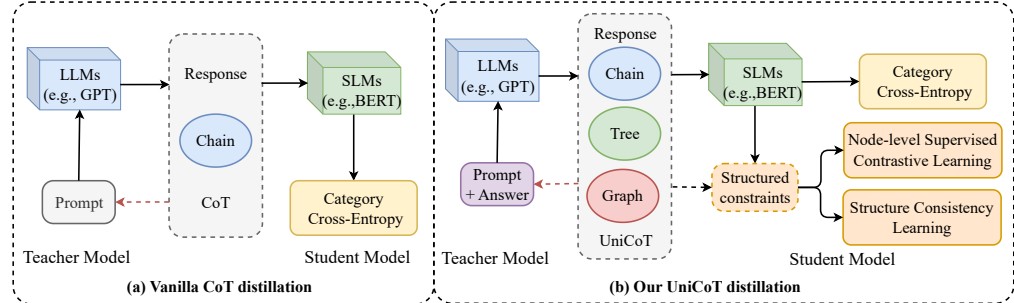

Figure 1: We compare different architectures: (a) generating vanilla CoT for distillation; (b) our UniCoTT, which uses UniCoT as a bridge to transfer knowledge between diverse structural thoughts.

2023). Inspired by this, some works explore more powerful reasoning with structural constraints in LLMs, namely structural CoTs, such as CoT-SC (multi-chain) (Wang et al., 2022b), ToT (tree) (Yao et al., 2023), and GoT (graph) (Besta et al., 2023). However, current distillation methods (Li et al., 2023) ignore the knowledge transfer of structured reasoning, specifically the integration of structural CoTs for enhanced reasoning, in the process of distilling reasoning capabilities into SLM. While SCOTT (Wang et al., 2023) attempts to introduce answers in prompts to ensure consistency in the reasoning of LLMs, it fails to account for CoTs with diverse structures. In summary, this leaves behind two core challenges: (1) how to efficiently transfer the CoT capabilities that LLMs possess to SLMs while preventing LLMs from producing erroneous reasoning explanations that could degrade performance; (2) how to uniformly consider prompts with different structures, such as chain, tree, and graph structures, in the process of knowledge transfer (i.e., prompting LLMs and training SLMs).

To address these two challenges, in this paper, we propose a novel unified teacher-student distillation framework termed **UniCoTT** for transferring knowledge of diverse structural CoTs from LLMs to SLMs. UniCoTT tackles challenges through innovative strategies in the following two aspects:

(1) **UniCoTT prompts LLMs to construct precise structured CoTs (i.e., UniCoT) in a unified way.** UniCoTT combines answers to prompt LLMs for the accurate generation with structured thoughts, which we refer to as UniCoT. We conceptualize UniCoT as a pivotal bridge facilitating knowledge transfer within teacher-student models. Specifically, we iteratively generate diverse structured thoughts, subsequently achieving uniform representations of these reasoning explanations.

(2) **UniCoTT utilizes the proposed unified supervised learning and structural consistency learning strategies to transfer knowledge of structured CoT to SLMs.** Specifically, we first consider the explanations in UniCoT as nodes and then introduce node-level supervised contrastive learning to enhance supervised representations of SLMs based on the traditional cross-entropy loss. In addition, we propose a novel structural consistency learning approach to ensure that the hidden states output by SLMs satisfies the structural constraints of UniCoT during distillation. Structural consistency learning is achieved by minimizing the upper bound of the structural error between the state encoding output of SLMs and the optimal structural representation.

Through the proposed UniCoT modeling and structural optimization methods, UniCoTT effectively extends the reasoning capability of LLMs obtained by structural CoT to SLMs. To the best of our knowledge, we are the first to consider structured CoT in a unified manner in this task. Extensive experiments on multiple datasets of factual reasoning, multi-choice question answering and NLU tasks demonstrate the effectiveness and universality of UniCoTT.

In general, our contributions are three-fold: (1) We propose a unified iterative method to prompt LLMs to construct diverse structured CoTs. (2) We propose a novel structural consistency learning strategy to transfer structured knowledge into SLMs efficiently. (3) Extensive experiments on multiple datasets demonstrate the effectiveness and universality of UniCoTT.

## 2 RELATED WORK

**Prompted CoT Models**. LLMs have demonstrated the ability to master various tasks with minimal instruction, requiring only a few examples as prompts (Brown et al., 2020; Nye et al., 2021; Zhuang

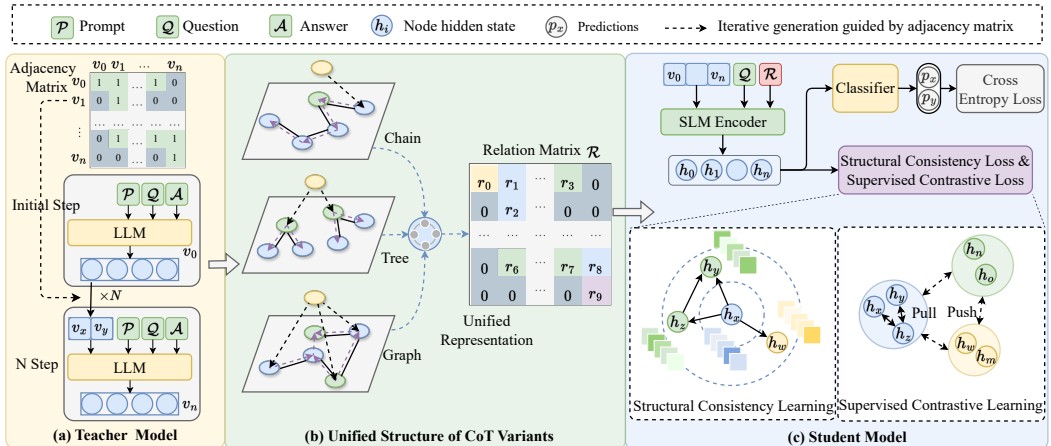

Figure 2: The illustration of our method, consisting of (1) LLMs construct UniCoT consisting of explanations with structural logic through N-step iteration; (2) SLMs obtain structured knowledge in UniCoT through node-level supervised contrastive loss and structural consistency learning.

et al., 2024e;f; 2025a). Furthermore, these models are capable of answering complex questions through the generation of sequential, step-by-step reasoning processes, known as Chains of Thought (CoT), even with few or no demonstrations (Wei et al., 2022b;a). However, the effectiveness of this technique is only reflected in extremely large LMs (with over 100B parameters) (Wei et al., 2022b;a), necessitating substantial computational resources or the need for costly API calls (Shridhar et al., 2022). Meanwhile, LLMs are prone to hallucinate unintended text (Ji et al., 2022) and produce illogical explanations (Wang et al., 2022a; Ye & Durrett, 2022). In this work, we aim to guide LLMs to generate rational explanations with diverse reasoning structures and transfer the reasoning knowledge of LLMs into SLMs.

**Knowledge Distillation**. The knowledge distillation method has been widely used in fields such as spoken understanding Zhuang et al. (2024a;c) and multimodal understanding Xie et al. (2024); Zhuang et al. (2024b;d; 2025b); Zhu et al. (2024); Yu et al. (2024). Our approach uses a teacher-student model to transfer knowledge from LLMs to SLMs, similar to a traditional distillation framework (Hinton et al., 2015). West et al. (2021) proposed a method for training the student model to complete knowledge gaps. Chan et al. (2022) introduces a strategy for learning a student model capable of making predictions solely based on a teacher model enhanced with fundamental principles. Shridhar et al. (2022) advocated for the training of student models to pose and answer subquestions. And Li et al. (2022) suggested training the student model on a dual task of generating both answers and rationales. Wang et al. (2023) utilizes a contrastive learning methodology to generate consistent explanations and fine-tune the student network using counterfactual reasoning. LI et al. (2022); Ho et al. (2022); Fu et al. (2023) also explored the chain CoT distillation method. In contrast, we use a structured CoT, i.e., UniCoT, to perform knowledge transfer.

**Structured CoT prompt learning**. Current studies indicate that human reasoning processes involve complex mechanisms of backtracking and feedforward logic (Sloman, 1996; Yao et al., 2023). In the exploration of novel ideas, humans develop intricate networks of thought, such as tree or graph reasoning paths (Dougherty & Franco-Watkins, 2000; Besta et al., 2023). Aligning with the complex reasoning of humans, some works explore more powerful reasoning with structural constraints in LLMs, namely structural CoTs, such as CoT-SC (Wang et al., 2022b), ToT (Yao et al., 2023), and GoT (Besta et al., 2023).

## 3 METHODS

In this section, we first introduce the overview of our UniCoTT similar to a teacher-student architecture in Section 3.1. We conceptualize UniCoT as a pivotal bridge facilitating knowledge transfer and describe methods for constructing UniCoT in Section 3.2. Subsequently, we introduce the proposed unified supervised learning strategy in Section 3.3 and the structural consistency learning approach in Section 3.4, aimed at facilitating supervised learning and representation learning for structural knowledge transfer, respectively. Finally, we summarize our total training objective in Section 3.5.

Figure 3: The illustration of CoTs with different structures and their corresponding relation matrices. We propose to construct diverse structures of CoTs in a unified way and use relation matrices to represent their structural constraints uniformly.

## 3.1 OVERVIEW OF UNICOTT

Our goal is to (1) prompt LLMs to explore various structured reasoning pathways for thoughts, as illustrated in Figure 3, and summarize these into unified representations (i.e., UniCoT) serving as bridges for knowledge transfer, and subsequently, (2) efficiently transfer semantic knowledge from UniCoT to small student models through unified supervision and representation learning.

UniCoTT is generally divided into two principal parts, namely the teacher network and the student network, where the teacher model is a network with an LLM as the base model. In this work, we focus on question-answer (QA) and natural language understanding (NLU) tasks. Specifically, we standardize QA and NLU tasks to be a universal setup: given a question or description $q$, models are tasked with predicting the gold answer or classification label $a^*$. This approach aligns with settings adopted in prior research (Wang et al., 2023).

**Teacher Network.** Similar to Wang et al. (2023), we utilize in-context learning to prompt teacher networks to obtain explanations for each question-answer pair $< q, a^* >$. Specifically, we employ a few annotated examples as demonstrations and prompt LLMs to generate an initial explanation $v_0$ for $< q, a^* >$. Subsequently, we iterate this process to prompt LLMs to produce the next explanation node $v_t$ given $< q, a^* >$ and a set of prefix explanation $\mathcal{P}(v_i) = [v_0, \cdots v_{i-1}]$. Therefore, each optimal explanatory node $v_i$ will be generated in the following unified way:

$$v_i^* = \arg\max \log P\left(v_i \mid p, q, a^*, \mathcal{P}(v_i)\right), \tag{1}$$

where $p$ denotes an input prompt. We further define all explanations generated by LLMs as sets of thoughts $\mathcal{T}$. Different from SCOTT which only produces chain explanation, we propose generating explanations with diverse reasoning structures as discussed in Section 3.2.

**Student Network.** We employ SLMs as the student, which leverage a series of explanations (i.e., UniCoT) generated by LLMs for learning. One straightforward implementation is fine-tuning the student network over the explanatory sets $\mathcal{T}$ and questions $q$ generated by the teacher. We can fine-tune the student using the standard categorical cross-entropy loss as:

$$\mathcal{L}_{cce}(y) = \sum_i a_i^* \log(\hat{y}_i), \tag{2}$$

where $\hat{y}_i$ denotes the predictions and $a_i^*$ denotes the corresponding label or answer. Our UniCoT comprises a set of thoughts characterized by a specific reasoning structure. To facilitate efficient learning of structured explanations by students, we further propose a structural consistency learning strategy to train SLMs as discussed in Section 3.4.

## 3.2 CONSTRUCTION OF UNICOT

Unlike previous works such as SCOTT (Wang et al., 2023), we consider the structural information during distillation and propose a novel prompt strategy to encourage LLMs to generate structured thoughts (i.e., chain, tree, and graph) that correspond to human reasoning processes, and unify them as UniCoT. A specific instantiation of UniCoT involves two key steps: (1) Iteratively constructing different structures of thoughts, and (2) Unifying the representation of different structures of thoughts.

**Iteratively Constructing UniCoT.** As shown in Figure 2(a), we initially create an adjacency matrix $\mathcal{A} = \{a_{ij}\} \in \mathbb{R}^{N_v \times N_v}$ representing a directed graph, where $N_v$ is the number of nodes and $a_{ij}$ indicates the presence of a directed edge from node $v_i$ to node $v_j$. As shown in Figure 3, the

adjacency matrix $\mathcal{A}$ is initialized differently for each structural variant of UniCoTT. For chain- and tree-structured UniCoTT, $\mathcal{A}$ is configured as a simple linear chain and a three-layer binary tree. For graph-structured UniCoTT, we randomly assign multiple connections for each node while ensuring the overall graph remains connected. The explanation generation process leverages parent nodes' explanations as context, which are fed into the teacher model as prompts to generate explanations for the current node. Specifically, we provide a prompt $p_0$ and question-answer pair $< q, a^* >$ to LLMs to generate the initial node $v_0$.

Subsequently, we construct a set of prefix explanations $\mathcal{P}(v_1)$ pertaining to $v_0$ based on $\mathcal{A}$. Further, LLMs are prompted to procure a specific explanation for $v_1$: $v_1 \sim p_{\text{LLM}}(v_1|p, q, a^*, \mathcal{P}(v_1))$. This generation process is iteratively conducted similarly until the terminal node $v_{N_v}$ is produced as:

$$v_t \sim p_{\text{LLM}}(v_t|p, q, a^*, \mathcal{P}(v_t)), \qquad t \in [1, N_v]. \tag{3}$$

Through this iterative process, a comprehensive explanation set $\mathcal{V} = \{v_1, v_2, \ldots, v_{N_v}\}$ is formulated. More detailed processes and specific prompts are provided in **Appendix** A.3 and A.8.

**Unified Representation of UniCoT.** Utilizing the adjacency matrix $\mathcal{A}$, we can iteratively derive all explanations (nodes) of UniCoT, denoted as $\mathcal{V} = \{v_0, v_1, \cdots, v_{N_v}\}$. Additionally, we introduce a relation matrix $\mathcal{R} = \{r_{ij}\} \in \mathbb{R}^{N_v \times N_v}$ to represent the structural relationships among all explanations within UniCoT:

$$r_{ij} = \frac{a_{ij}}{\text{Dij}(v_i, \phi(v_i))}, \tag{4}$$

where, $\phi(v_i)$ denotes the nearest sink node to $v_i$ and $\text{Dij}(\cdot, \cdot)$ represents the shortest path between two nodes determined by the Dijkstra algorithm.

The construction process of UniCoT reveals that the closer the explanation is to the sink node, the more refined it becomes, thereby semantically nearing the answer. Consequently, as illustrated in the Figure 3, the values within $\mathcal{R}$ serve a dual purpose: (1) they represent the connectivity relations between nodes within UniCoT, and (2) concurrently quantify the proximity of nodes to the answer.

## 3.3 NODE-LEVEL SUPERVISED CONTRASTIVE LOSS

To further leverage the explanations generated by LLMs for optimizing the student network, we incorporate node-level supervised contrastive learning based on our UniCoT within the fine-tuning phase. Given a collection of question-answer pairs $\{< q, a^* >\}$ and their corresponding explanations $\mathcal{V} = \{v_0, \cdots, v_{N_v}\}$ within UniCoT, We can derive the hidden state of nodes $\mathcal{H} = [h_0, \cdots, h_{N_v}]$ through the encoder of SLMs:

$$h_j = \text{Encoder}_{\text{SLM}}(v_j), \quad \forall j \in \{0, \ldots, N_v\}, \tag{5}$$

where $h_j$ represents the hidden state encoding for the $j$-th node $v_j$.

We further select positive samples from other nodes $v_{j'}$ with the same answer and negative samples $v_k^n$ from nodes with different labels. Utilizing these positive and negative samples, the node-level supervised contrastive loss is formulated as:

$$\mathcal{L}_{\text{nsc}} = - \sum_{j,j'=0, j!=j'}^{N_v} \log \frac{\exp(v_j \cdot v_{j'}/\tau)}{\sum_{k=1}^K \exp(v_j \cdot v_k^n/\tau)}, \tag{6}$$

where $\tau$ is the temperature parameters and $K$ denotes the number of negative samples. $\mathcal{L}_{\text{nsc}}$ can effectively minimize the distance between similar samples and maximize the separation between dissimilar samples, thereby enhancing the capability of SLM to recognize similar samples.

## 3.4 STRUCTURAL CONSISTENCY LEARNING

As shown in Figure 2, UniCoTT encourages the output of the SLM encoder to conform to the structured constraints of UniCoT. Structural constraints are grounded in the intuition that nodes within the same reasoning path should be proximal in the latent space, whereas nodes from divergent reasoning paths should be distanced. We propose a novel structural consistency learning strategy to optimize the student and maintain structural representations in a unified and effective manner.

**Structural Error and Upper Bound.** We first define the structured hidden state output by the SLM encoder as $\mathcal{S} = \mathcal{RH} = [s_0, \cdots, s_{N_v}]$. Assuming $\chi = \{\chi_0, \cdots \chi_{N_v}\}$ represents the optimal representation, We can quantify the structural error between $\mathcal{S}$ and the optimal representation as:

$$\xi_\chi = \sum_i^{N_v} \|\mathbf{W} \cdot s_i - \chi_i\|, \tag{7}$$

where $\mathbf{W}$ is a learnable mapping matrix designed to align $\mathcal{S}$ with $\chi$. However, the optimal representation $\chi$ is unknown a posterior, and it is difficult to directly optimize the student by minimizing $\xi_\chi$. To address this challenge, we theoretically derive an upper bound for $\xi_\chi$ in Theorem 1 based on findings (Shwartz-Ziv et al., 2023; Srinath Halvagal et al., 2024; Foo et al., 2023; Bardes et al., 2022).

**Theorem 1** *(**Upper Bound on Structural Error**) For any optimal representation $\chi$, the structural error $\xi_\chi$ as defined in Equation 7 is bounded above by the product of the Frobenius norms of the projection operator and the representation itself:*

$$\xi_\chi \leq \|\mathbf{T}_\mathcal{S}\|_F \|\chi\|_F = \|\mathbf{I} - \mathcal{S}^\top(\Sigma_\mathcal{S})^\dagger\mathcal{S}\|_F \|\chi\|_F, \tag{8}$$

*where, $\mathbf{T}_\mathcal{S} = \mathbf{I} - \mathcal{S}^\top(\Sigma_\mathcal{S})^\dagger\mathcal{S}$ be the orthogonal projection operator onto the complement of the span of $\mathcal{S}$, $\mathbf{I}$ is the identity matrix, $(\cdot)^\dagger$ is the pseudoinverse and $\Sigma_\mathcal{S} = \mathcal{S}^\top\mathcal{S}$ is the covariance matrix of $\mathcal{S}$.*

The proof of Theorem 1 is provided in Appendix A.2. Given that $\|\chi\|_F$ in Eq. 8 is a is a fixed but unpredictable quantity, we can minimize structural error $\xi_\chi$ by minimizing $\|\mathbf{T}_\mathcal{S}\|_F$. Based on previous theoretical findings (Shwartz-Ziv et al., 2023; Bardes et al., 2022), we can minimize $\|\mathbf{T}_\mathcal{S}\|_F$ by maximizing the rank of $\Sigma_\mathcal{S}$. UniCoTT achieves this by maximizing the diagonal term of Z with a structural decoupling loss $\mathcal{L}_{sd}$ and minimizing its off-diagonal term with a structural entanglement loss $\mathcal{L}_{se}$:

The structural decoupling loss $\mathcal{L}_{sd}$ maximizes the diagonal entries and encourages the student network to learn distinct and non-overlapping structural representations. $\mathcal{L}_{sd}$ is formulated as:

$$\mathcal{L}_{sd} = \frac{1}{D} \sum_{d=1}^{D} \text{ReLU}\left(1 - \sqrt{\sigma_d + \tau_{sd}}\right), \tag{9}$$

where $D$ represents the dimension of $\mathcal{S}$, $\text{ReLU}(\cdot)$ is the Rectified Linear Unit (ReLU) function, $\sigma_d$ is the variance of the $d$-th dimension across the vectors $s_0, \cdots, s_{N_v}$ and $\tau_{sd}$ is a temperature coefficient.

The structural entanglement loss $\mathcal{L}_{se}$ minimizes the off-diagonal entries of $\Sigma_\mathcal{S}$ and consequently minimizes the correlation between dimensions in the latent space. $\mathcal{L}_{se}$ is formulated as:

$$\mathcal{L}_{se} = \frac{1}{D \times (N_v - 1)} \sum_{i != j} \Sigma_\mathcal{S}[i, j]. \tag{10}$$

UniCoTT maintains structural constraints in the latent space during fine-tuning through structural consistency loss $\mathcal{L}_{sc} = \mathcal{L}_{sd} + \mathcal{L}_{se}$, thereby achieving efficient transfer of UniCoT knowledge.

## 3.5 OVERALL OBJECTIVE FOR THE STUDENT

**Training Objective.** Our overall loss comprises a category cross-entropy loss $\mathcal{L}_{cce}$, a node-level supervised contrastive loss $\mathcal{L}_{nsc}$ and a structural consistency loss $\mathcal{L}_{sc}$. Therefore, our fine-tuning loss can be expressed as:

$$\mathcal{L}_{total} = \mathcal{L}_{cce} + \alpha\mathcal{L}_{nsc} + \beta\mathcal{L}_{sc}, \tag{11}$$

where $\alpha$ and $\beta$ are trade-off hyperparameters.

**Inference.** We use the final output of SLMs to evaluate the effectiveness of knowledge transfer. Following previous work and implementation (Wang et al., 2023), student models can obtain explanations generated by CoTs at the inference stage of all methods including the baselines and our UniCoTT. In addition, to avoid leaking correct answers within CoT explanations during inference, we use "|MASK|" identifiers to mask both the answers (targets) and candidate answers in CoT explanations. Note that we employ the same operation for all baselines for fair comparison.

## 4 EXPERIMENTS

### 4.1 EXPERIMENTAL SETUP

**Datasets.** We conduct experiments across three types of tasks: (1) **Factual Reasoning Task.** We evaluate our UniCoTT over CREAK (Onoe et al., 2021), StrategyQA (Geva et al., 2021) and CSQA2 (Talmor et al., 2021) datasets, which contain a large number of questions about entity knowledge and commonsense reasoning. (2) **Multiple-Choice Question Answer.** We select CSQA (Talmor et al., 2018), QASC (Khot et al., 2020), and OBQA (Mihaylov et al., 2018) datasets to evaluate our method on multiple-choice comprehension question answering task. (3) **Natural Language Understanding (NLU).** In the realm of NLU, we utilized the CoLA (Warstadt et al., 2019), RTE (Poliak, 2020), MNLI (Williams et al., 2018), MRPC (Dolan & Brockett, 2005) datasets from GLUE benchmark (Wang et al., 2018) to evaluate the performance of UniCoTT.

Table 1: A performance comparison of various methods on the factual reasoning benchmark, with the best results emphasized in **bold**. The Base Model is the student network used for training. The results of using more pre-training language models as the base model are provided in Appendix A.4.

| Base Model | Method | Structure | CREAK | | | CSQA2 | | | StrategyQA | | |
|---|---|---|---|---|---|---|---|---|---|---|---|
| | | | Acc. | F1 | Ins. | Acc. | F1 | Ins. | Acc. | F1 | Ins. |
| BERT-base | +None | - | 69.3 | 69.1 | 70.1 | 55.1 | 55.0 | 55.4 | 82.7 | 82.7 | 82.3 |
| | CoT | Chain | 77.7 | 76.6 | 78.7 | 71.1 | 71.0 | 70.9 | 87.7 | 87.6 | 88.4 |
| | SCOTT | Chain | 84.1 | 84.2 | 83.5 | 85.2 | 85.2 | 86.6 | 90.0 | 89.8 | 90.8 |
| | DSbS | Chain | 69.5 | 69.5 | 69.4 | 54.2 | 54.1 | 54.9 | 81.0 | 80.9 | 80.7 |
| | UniCoTT | Chain | 92.7 | 92.8 | 93.0 | 81.5 | 81.4 | 81.8 | 90.9 | 90.9 | 91.1 |
| | UniCoTT | Tree | 94.5 | 94.4 | 94.9 | **87.9** | **87.9** | **89.2** | **93.4** | **93.5** | **94.0** |
| | UniCoTT | Graph | **95.8** | **95.6** | **96.0** | 83.8 | 83.4 | 84.9 | 92.1 | 92.4 | 93.2 |
| RoBERTa-base | +None | - | 71.3 | 71.3 | 71.4 | 56.0 | 55.8 | 55.7 | 83.9 | 83.9 | 84.1 |
| | CoT | Chain | 86.5 | 86.4 | 86.7 | 72.7 | 72.6 | 72.4 | 86.6 | 86.5 | 90.0 |
| | SCOTT | Chain | 90.2 | 90.2 | 90.5 | 82.3 | 82.3 | 81.6 | 91.5 | 91.2 | 90.9 |
| | DSbS | Chain | 72.2 | 72.2 | 72.4 | 54.2 | 54.0 | 55.3 | 80.0 | 80.0 | 80.2 |
| | UniCoTT | Chain | 93.4 | 93.4 | 93.3 | 82.2 | 82.6 | 82.0 | 93.6 | 93.4 | 94.4 |
| | UniCoTT | Tree | 94.8 | 94.6 | 94.7 | **88.8** | **88.9** | **90.2** | **94.6** | **94.6** | **95.5** |
| | UniCoTT | Graph | **96.8** | **96.8** | 95.9 | 84.9 | 84.6 | 85.9 | 94.2 | 93.9 | 94.7 |

Table 2: A performance comparison of various methods on the Multiple-Choice QA benchmark, with the best results emphasized in **bold**. The Base Model is the student network used for training. The results of using more pre-training language models as the base model are provided in Appendix A.4.

| Base Model | Method | Structure | CSQA | | | OBQA | | | QASC | | |
|---|---|---|---|---|---|---|---|---|---|---|---|
| | | | Acc. | F1 | Ins. | Acc. | F1 | Ins. | Acc. | F1 | Ins. |
| BERT-base | +None | - | 81.6 | 68.6 | 57.4 | 75.9 | 65.7 | 52.8 | 84.8 | 55.8 | 24.2 |
| | CoT | Chain | 86.7 | 77.0 | 71.1 | 77.5 | 69.0 | 61.4 | 89.3 | 73.6 | 57.0 |
| | SCOTT | Chain | 88.7 | 80.2 | 77.3 | 80.8 | 71.0 | 64.4 | 86.4 | 64.5 | 53.8 |
| | DSbS | Chain | 81.3 | 68.2 | 56.2 | 76.4 | 63.2 | 52.0 | 87.1 | 54.5 | 22.8 |
| | UniCoTT | Chain | 88.1 | 80.9 | 79.2 | 82.4 | 75.8 | 73.6 | 92.3 | 81.1 | 70.3 |
| | UniCoTT | Tree | 90.4 | 84.9 | 84.4 | 83.8 | 75.6 | 75.2 | **93.2** | **83.3** | **77.8** |
| | UniCoTT | Graph | **91.6** | **88.0** | **86.8** | **84.4** | **77.9** | **77.2** | 90.3 | 83.6 | 68.5 |
| RoBERTa-base | +None | - | 83.3 | 72.6 | 64.4 | 78.4 | 69.5 | 61.4 | 86.7 | 58.5 | 32.3 |
| | CoT | Chain | 86.7 | 77.7 | 71.0 | 84.4 | 78.3 | 74.4 | 90.9 | 75.4 | 60.3 |
| | SCOTT | Chain | 89.8 | 83.7 | 78.9 | 84.8 | 77.9 | 73.0 | 87.5 | 60.4 | 30.8 |
| | DSbS | Chain | 82.8 | 72.4 | 64.3 | 77.6 | 69.0 | 60.0 | 87.5 | 59.7 | 34.6 |
| | UniCoTT | Chain | 91.7 | 86.5 | 86.7 | 84.3 | 79.2 | 78.7 | 92.9 | 84.6 | 78.6 |
| | UniCoTT | Tree | 91.7 | 86.9 | 87.5 | 87.5 | 83.4 | 82.2 | **93.6** | **84.5** | **80.1** |
| | UniCoTT | Graph | **92.5** | **89.6** | **88.8** | **88.8** | **85.4** | **84.1** | 92.4 | 83.7 | 75.7 |

**Evaluation Metrics.** Following the previous work settings (Wang et al., 2023), we adopt accuracy (Acc.), F1 score (F1), and instance accuracy (Ins.) to measure the performance of the model in factual reasoning and multiple-choice question-answering task. In addition, we adopt Matthews correlation

coefficients (Mcc.) to evaluate the performance of the model on the CoLA dataset and use accuracy to evaluate the performance of the model on the RTE, MNLI, and MRPC datasets.

**Baselines.** We compare our approach with the small pre-trained language models (PLMs), CoT (Wei et al., 2023), SCOTT (Wang et al., 2023) and DSbS (Hsieh et al., 2023). (1) PLMs: We utilize BERT (Devlin et al., 2019), RoBERTa (Liu et al., 2019), and XLNet (Yang et al., 2019) as the backbones, and directly fine-tune them as baselines, which are fine-tuned directly to serve as baselines for comparison. (2) CoT: We use the CoT technique to generate the reasoning contents of the model and then distill them on the small PLMs. (3) SCOTT: We utilize the SCOTT (Wang et al., 2023) framework to generate the reasoning content of the model through counterfactual reasoning and compare it with our UniCoTT.

**Implementation Details.** We use the LLM `gpt-3.5-turbo-1106` (Ouyang et al., 2022) as the teacher network for the construction of CoT and UniCoT. We use RoBERTa (Liu et al., 2019), BERT (Devlin et al., 2019), and XLNet (Yang et al., 2019) as base models for our student network. The hyperparameter $\alpha$ and $\beta$ in Eq. 11 are set to 0.5 and 0.2 to achieve the optimal performance in experiments. The hidden size for text is set to 768. We employ Adam as the optimizer with a weight decay of 0.01. We will publish the complete code after the paper is accepted. More training details and results can be found in Appendix A.

Table 3: A performance comparison on the GLUE benchmark, with the best results emphasized in **bold**. I and II indicates using BERT-base and RoBERTa-base as the base model, respectively.

| Base Model | Methods | Structure | CoLA | RTE | WNLI | MRPC | Average |
|---|---|---|---|---|---|---|---|
| | +None | - | 56.6 | 65.3 | 53.4 | 81.8 | 64.3 |
| | CoT | Chain | 67.9 | 81.6 | 80.3 | 87.8 | 79.4 |
| | SCOTT | Chain | 81.1 | 91.7 | 91.6 | 92.6 | 89.3 |
| BERT-base | UniCoTT | Chain | 86.4 | 89.9 | 93.0 | 95.5 | 91.2 |
| | UniCoTT | Tree | 88.5 | 93.5 | **94.4** | **96.3** | **93.2** |
| | UniCoTT | Graph | **90.2** | **94.6** | 93.9 | 94.1 | **93.2** |
| | +None | - | 56.7 | 78.7 | 55.5 | 86.9 | 69.5 |
| | CoT | Chain | 69.6 | 82.3 | 81.2 | 71.3 | 76.1 |
| | SCOTT | Chain | 78.5 | 89.9 | 90.8 | 90.6 | 87.5 |
| RoBERTa-base | UniCoTT | Chain | 88.3 | 91.7 | 93.4 | 93.1 | 91.6 |
| | UniCoTT | Tree | 91.4 | 93.0 | 95.1 | **96.3** | 94.0 |
| | UniCoTT | Graph | **93.9** | **95.5** | **95.3** | 94.0 | **94.7** |

## 4.2 MAIN RESULTS

**Evaluation on Factual Reasoning Tasks.** We present our experimental results on three datasets in Table 1. Analysis of Table 1 yields several insights: **(1)** Enriching PLMs with additional explanations including CoT, SCOTT and UniCoTT significantly enhances their performance in factual reasoning tasks. We attribute this to the incorporation of more prior knowledge within explanations into PLMs. **(2)** Our method of employing solely chain UniCoT achieves superior performance compared to other baselines that transfer knowledge through chain explanation. The performance of both CoT and SCOTT is surpassed by our method of employing chain UniCoT. This disparity may stem from the improvement of representation by structured constraints and node-level self-supervised contrastive learning of our UniCoTT. **(3)** We can observe that different reasoning structures have different performance on different datasets. Tree UniCoT and graph UniCoT with more complex reasoning structures have a higher performance improvement than chain UniCoT. In addition, our method outperforms all baselines in terms of PLM performance gains across all backbone networks. It demonstrates the effectiveness and generalization of our UniCoTT in transferring LLMs knowledge.

**Evaluation on Multiple-Choice QA Tasks.** As illustrated in Table 2, we evaluate the performance of our method on CSQA, OBQA, and QASC datasets. where we have several detailed observations:

**(1)** The improvement in model performance by incorporating SCOTT or vanilla CoT is not as significant as the gain by incorporating our UniCoT. This further indicates that considering different reasoning structures is of great significance for improving the comprehension capabilities of SLMs. **(2)** Various reasoning structures manifest distinct performances across complex commonsense question-

answering datasets. The graph UniCoT exhibits superior results in CSQA and OBQA, while the tree UniCoT performs better in QASC. This further demonstrates that our approach offers a twofold advantage: it unifies diverse thought structures and significantly improves performance. **(3)** We can further observe that the performance of SCOTT is lower than that of the base model when using RoBERTa-base as the backbone on the QASC dataset. This indicates providing unreasonable explanations can adversely impact the representations and comprehension capabilities of SLMs.

**Evaluation on NLU Tasks.** To evaluate the universality of our UniCoTT, we conduct experiments utilizing four datasets from the GLUE benchmark, as result shown in Table 3. The results demonstrate that our model consistently improves the performance of BERT and RoBERTa across various metrics on text classification tasks. Furthermore, we observe that the performance of the student remains affected by reasoning structures in NLU tasks. This indicates the necessity of optimizing structural constraints and representation during fine-tuning.

Table 4: The ablation experiment results of UniCoTT on the CREAK, OBQA and QASC datasets, with the best results emphasized in **bold**.

| Structure | Method | CREAK | | | OBQA | | | QASC | | |
|---|---|---|---|---|---|---|---|---|---|---|
| | | Acc. | F1. | Ins. | Acc. | F1. | Ins. | Acc. | F1. | Ins. |
| Chain | w/o $\mathcal{L}_{nsc}$ in Eq. 6 | 92.3 | 92.2 | 92.0 | 82.5 | 77.6 | 76.3 | 91.5 | 82.9 | 73.1 |
| | w/o $\mathcal{L}_{sd}$ in Eq. 9 | 91.8 | 91.8 | 91.7 | 81.2 | 78.1 | 75.2 | 90.6 | 80.5 | 71.4 |
| | w/o $\mathcal{L}_{se}$ in Eq. 10 | 92.5 | 92.3 | 91.8 | 82.8 | 77.6 | 77.0 | 91.2 | 83.3 | 75.5 |
| | Full UniCoTT Method | **93.4** | **93.4** | **93.3** | **84.3** | **79.2** | **78.7** | **92.9** | **84.6** | **78.6** |
| Tree | w/o $\mathcal{L}_{nsc}$ in Eq. 6 | 93.1 | 93.0 | 92.5 | 86.2 | 81.9 | 81.8 | 91.6 | 83.1 | 77.7 |
| | w/o $\mathcal{L}_{sd}$ in Eq. 9 | 92.3 | 92.3 | 92.0 | 85.5 | 81.1 | 79.5 | 90.4 | 80.1 | 72.2 |
| | w/o $\mathcal{L}_{se}$ in Eq. 10 | 93.5 | 93.4 | 93.6 | **88.3** | **83.9** | **83.0** | 91.3 | 82.7 | 76.3 |
| | Full UniCoTT Method | **94.8** | **94.6** | **94.7** | 87.5 | 83.4 | 82.2 | **93.6** | **84.5** | **80.1** |
| Graph | w/o $\mathcal{L}_{nsc}$ in Eq. 6 | 93.5 | 93.5 | 93.0 | 86.8 | 82.3 | 81.5 | 91.0 | 79.9 | 72.5 |
| | w/o $\mathcal{L}_{sd}$ in Eq. 9 | 93.2 | 93.0 | 93.2 | 86.0 | 81.7 | 80.2 | 90.2 | 77.5 | 67.7 |
| | w/o $\mathcal{L}_{se}$ in Eq. 10 | 95.0 | 95.1 | 94.9 | 88.2 | 84.7 | 83.5 | 91.4 | 81.9 | 72.4 |
| | Full UniCoTT Method | **96.8** | **96.8** | **95.9** | **88.8** | **85.4** | **84.1** | **92.4** | **83.7** | **75.7** |

## 4.3 ABLATION STUDY AND ANALYSIS

**The Effect of the Node-level Supervision Contrastive Loss.** We employ RoBERTa-base as the backbone and conduct experiments to study the impact of our node-level supervised contrastive loss as illustrated in Table 4. The results reveal a significant decrease in performance across all evaluated structures when the node-level supervised contrastive loss is not employed to refine the representations of SLMs. This verifies the following: (1) $\mathcal{L}_{nsc}$ is capable of aggregating priori information provided by LLMs sufficiently and effectively, and (2) the refinement of explanation features in latent space plays a significant role in enhancing the expressive capabilities of SLMs.

**Quantitatively Evaluate the Rationality of the Explanations.** To evaluate the consistency between the rationales generated by the teacher and the gold answers, we use the LAS metric (Hase et al., 2020; Wang et al., 2023), whose core idea is to measure how well the rationales assist a simulator to predict the gold answers. We adopt the same settings following (Hase et al., 2020; Wang et al., 2023) to perform evaluation on datasets CSQA, OBQA and QASC, with

Table 5: Comparison with different methods using the LAS metric (Hase et al., 2020; Wang et al., 2023) to evaluate the output rationality.

| Methods | Structures | CSQA | OBQA | QASC |
|---|---|---|---|---|
| CoT | Chain | 3.4 | 6.0 | 4.2 |
| SCOTT | Chain | 6.5 | 6.4 | 0.8 |
| UniCoTT | Chain | 8.4 | 5.9 | 6.2 |
| UniCoTT | Tree | 8.4 | 9.1 | **6.9** |
| UniCoTT | Graph | **9.2** | **10.4** | 6.4 |

the result as Figure 5. Higher LAS values represent higher inference consistency, which is computed as the difference between the task performance when the rationale is provided as input $vs.$ when it is not. The results show the consistency between the rationales generated by teachers and the gold answers measured by LAS, which verifies that UniCoTT can ensure the rationality of explanations.

**The Effect of the Structural Consistency Learning.** To evaluate the impact of our structural consistency learning strategy on model performance, we remove $\mathcal{L}_{sd}$ and $\mathcal{L}_{se}$, respectively. We

|  | Factual Reasoning | | | Multi-choice QA | | Mathematical Reasoning |
|  | CREAK | CSQA2 | StrategyQA | CSQA | OBQA | GSM8K |
|---|---|---|---|---|---|---|
| Base | 88.8 | 63.7 | 83.2 | 92.0 | 91.0 | 76.9 |
| UniCoTT (Ours) | 91.5 | 75.4 | 88.7 | 95.0 | 92.9 | 79.2 |

Table 6: Performance comparison on the factual reasoning, multi-choice QA and mathematical reasoning benchmarks (i.e., GSM8K Cobbe et al. (2021)). We use `Qwen2.5-3B-Instruct` as base model for SFT with LoRA. The more details and loss curve are provided in Appendix A.9.

present our experimental results on three different types of datasets in Table 4. The backbone models use `RoBERTa-base`. Observations indicate that the absence of structural decoupling loss or structural entanglement loss significantly diminishes performance. This suggests that structural reasoning information within UniCoT is essential in transferring knowledge to SLMs. Additionally, this result corroborates the effectiveness of our structural consistency learning in harmonizing structural reasoning information and enhancing the expressive capabilities of SLMs.

## 4.4 EVALUATION WITH DECODER-ONLY MODELS

To further evaluate the efficacy of our approach, we employed the decoder-only `Qwen2.5-3B-Instruct` (Team, 2024) as our foundation model for conducting experiments. However, popular decoder-only architectures typically adhere to the next-token prediction paradigm. The predictive nature of such generative models presents a challenge in directly implementing our designed structural constraints for classification. Consequently, we modified our training methodology for structured CoT to accommodate these decoder-only models while preserving our unified structural CoT distillation capabilities. Specifically, we utilized our generated graph-based UniCoT as instruction input for `Qwen2.5-3B-Instruct` (Team, 2024) and incorporated our structural constrained adjacency matrix as additional prompts into the decoder-only model architecture. The model training still adhered to the next-token prediction paradigm via supervised fine-tuning training (SFT) with low-rank adaptation (LoRA) (Hu et al., 2022). Through this approach, we maintained the advantages of unified distillation of structured CoT while ensuring compatibility with decoder-only architectures. Please refer to our appendix for more details.

As shown in Table 6, our proposed UniCoTT method demonstrates consistent improvements across various reasoning tasks compared to the base model, highlighting its effectiveness for decoder-only architectures. In factual reasoning, UniCoTT achieves gains of 2.7%, 11.7%, and 5.5% on CREAK, CSQA2, and StrategyQA respectively, with CSQA2 showing the most significant improvement. For multi-choice question answering, we observe increases of 3.0% on CSQA and 1.9% on OBQA. In mathematical reasoning, UniCoTT improves performance on GSM8K by 2.28%. These results illustrate UniCoTT's robust performance enhancement across diverse reasoning tasks, with particularly notable gains in complex factual reasoning.

## 5 CONCLUTION

In this paper, we present a unified distillation framework, UniCoTT, designed for CoT with diverse reasoning structures. We propose an efficient method for generating CoT with diverse structures and its unified representation approach. In addition, we introduce a node-level supervised contrastive loss and structural consistency learning strategy, aimed at facilitating supervised learning and representation learning for structural knowledge transfer, respectively. In theory, we derive an upper bound for the structural representation error and achieve structural constraints by optimizing this upper bound. The experimental results show that UniCoTT can effectively improve the performance of SLMs on factual reasoning, multiple-choice QA, and NLU tasks.

**Limitations.** The construction of UniCoT relies on APIs of LLMs, which may not be easy to implement in specific situations. Therefore, exploring more efficient and low-resource methods is the direction of future research in this article. In addition, this article studies factual reasoning, open-domain multiple-choice question answering and natural language understanding tasks, and further research can be conducted in more fields to evaluate the generality of our method in the future.

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

# A APPENDIX

## A.1 MORE TRAINING DETAILS

The constant $\tau$ and $\tau_{sd}$ for the tolerance of the intra-class variance are set as 0.1 and 0.2 to achieve the optimal performance in experiments. We tune all models for 6 epochs and set the learning rate of 3e-6 on all the datasets. When conducting experiments in CSQA and CSQA2, batch sizes were set to 5 and 2, respectively. In addition, we uniformly set the batch size for training on other datasets to 8. The effectiveness of our approach depends on two key hyperparameters: $\alpha$ and $\beta$, which control the balance between supervised learning components (supervised contrastive learning and cross-entropy) and structural constraints. To determine optimal values for these parameters, we employed a systematic grid search methodology. We first conducted a grid search for $\alpha$ within the range $[0.1, 0.9]$ on the CREAK dataset, which yielded an optimal value of $\alpha = 0.5$. Subsequently, with $\alpha$ fixed at 0.5, we performed a second grid search for $\beta$ within the same range $[0.1, 0.9]$, identifying $\beta = 0.2$ as the optimal value. These hyperparameter values, derived from the CREAK dataset, were then applied consistently across all other datasets in our experiments. Following the original paper, we use the gpt-neox-20b backbone to implement SCOTT. We implement all methods based on Huggingface Transformers (Wolf et al., 2020).

All experiments using the encoder-only models are conducted on 8 RTX 3090 GPUs. While our primary experiments focus on encoder-based models, we also extend UniCoTT to decoder-only architectures to validate its generalizability. The experiments using the decoder-only model (i.e., `Qwen2.5-3B-Instruct`) are conducted on A100-80G GPUs, and more details and settings regarding the use of a decoder only model for conducting experiments are specifically introduced in Section A.9.

## A.2 PROOF OF THEOREM 1

*Proof.* The optimization objective of minimizing structural error can be obtained from Eq. 7 as:

$$\mathbf{W} = \underset{\mathbf{W}'}{\text{minimize}} \, \|\mathbf{W}'\|_F \,, \tag{12}$$

$$\text{s.t. } \mathbf{W}' \in \underset{\hat{\mathbf{W}}}{\arg\min} \sum_{i}^{N_v} \|\mathbf{W} \cdot s_i - \chi_i\|^2 \,. \tag{13}$$

To solve for $\mathbf{W}$, we first define the vectorization: To tackle the problem of determining the matrix $\mathbf{W}$, we initiate by converting $\mathbf{W}$ into a vector form through the process of vectorization. This involves stacking all the elements of $\mathbf{W}$ into a single column vector $\mathbf{w}$:

$$\mathbf{w} = \text{vec}(\mathbf{W}) = \begin{bmatrix} \mathbf{w}_1 \\ \mathbf{w}_2 \\ \vdots \\ \mathbf{w}_{N_v} \end{bmatrix} \in \mathbb{R}^{N_v N_v}, \tag{14}$$

This allows us to represent the multiplication in terms of vector operations:

$$\mathbf{W}\mathbf{s}_i = \left(s_i^\top \otimes \mathbf{I}\right)\mathbf{w} = \tilde{\mathbf{S}}_i\mathbf{w}, \tag{15}$$

where, $\otimes$ is the Kronecker product and $\mathbf{I}$ is the identity matrix. Then, we obtain our optimization objective as:

$$\begin{aligned} f(\mathbf{W}) &= \sum_{i=1}^{N_v} \|\mathbf{W} \cdot s_i - \chi_i\|^2 \\ &= \sum_{i=1}^{N_v} \left\| \chi_i - \tilde{\mathbf{S}}_i \mathbf{w} \right\|^2 . \end{aligned} \tag{16}$$

Therefore, this optimization problem is essentially a convex optimization problem, and we can obtain a closed-form solution as follows:

$$\mathbf{W} = \chi^\top \mathcal{S}^\top \left(\mathcal{S}\mathcal{S}^\top\right)^\dagger = \chi^\top \mathcal{S}^\top \left(\Sigma_\mathcal{S}\right)^\dagger . \tag{17}$$

Substituting this into the downstream structured error:

$$\begin{aligned} \xi_\chi &= \frac{1}{N_v} \sum_{i=1}^{N_v} \|\mathbf{W} \cdot s_i - \chi_i\| \\ &= \frac{1}{N_v} \sum_{i=1}^{N_v} \sqrt{\sum_{r=1}^{R} \left((\mathbf{W} \cdot s_i)\,[r] - \chi_i[r]\right)^2} \end{aligned} \tag{18}$$

The Cauchy-Schwarz inequality is applied to bound the Euclidean norm of a sum of square roots by the square root of the sum of squares:

$$\frac{1}{N_v} \sum_{i=1}^{N_v} \sqrt{\sum_{r=1}^{R} \left((\mathbf{W} \cdot s_i)\,[r] - \chi_i[r]\right)^2} \leq \sqrt{\frac{1}{N_v} \sum_{i=1}^{N_v} \sum_{r=1}^{R} \left((\mathbf{W} \cdot s_i)\,[r] - \chi_i[r]\right)^2} \tag{19}$$

We use the standard norm notation to represent the formula inside the square root:

$$\sqrt{\frac{1}{N_v} \sum_{i=1}^{N_v} \sum_{r=1}^{R} \left( (\mathbf{W} \cdot s_i)[r] - \chi_i[r] \right)^2} = \frac{1}{\sqrt{N_v}} \left\| \mathbf{W} \mathcal{S} - \chi_{\mathcal{S}}^{\top} \right\|_F$$

$$= \frac{1}{\sqrt{N_v}} \left\| \chi^{\top} \mathcal{S}^{\top} (\Sigma_{\mathcal{S}})^{\dagger} \mathcal{S} - \chi_{\mathcal{S}}^{\top} \right\|_F \tag{20}$$

$$= \frac{1}{\sqrt{N_v}} \left\| \chi^{\top} \left( \mathcal{S}^{\top} (\Sigma_{\mathcal{S}})^{\dagger} \mathcal{S} - \mathbf{I} \right) \right\|_F$$

$$= \frac{1}{\sqrt{N_v}} \left\| \left( \mathbf{I} - \mathcal{S}^{\top} (\Sigma_{\mathcal{S}})^{\dagger} \mathcal{S} \right) \chi \right\|_F.$$

We further define the projection matrix $\mathbf{T}_{\mathcal{S}} = \mathbf{I} - \mathcal{S}^{\top} (\Sigma_{\mathcal{S}})^{\dagger} \mathcal{S}$, we obtain our upper bound as :

$$\xi_\chi \leq \frac{1}{\sqrt{N_v}} \left\| \left( \mathbf{I} - \mathcal{S}^{\top} (\Sigma_{\mathcal{S}})^{\dagger} \mathcal{S} \right) \chi \right\|_F$$

$$\leq \frac{1}{\sqrt{N_v}} \left\| \mathbf{T}_{\mathcal{S}} \chi \right\|_F \tag{21}$$

$$\leq \left\| \mathbf{T}_{\mathcal{S}} \chi \right\|_F \leq \left\| \mathbf{T}_{\mathcal{S}} \right\|_F \left\| \chi \right\|_F,$$

which concludes the proof for Theorem 1.

Based on Theorem 1 and existing theoretical findings (Shwartz-Ziv et al., 2023; Srinath Halvagal et al., 2024; Foo et al., 2023; Bardes et al., 2022), we can reduce structural error by minimizing the upper bound of structural error.

### A.3 More details of the algorithm

To demonstrate the construction process of our UniCoT more clearly, we provide algorithmic pseudocode for uniformly constructing different structures of CoT, namely UniCoT, in Alg. 1 and Code 1.

The adjacency matrix $A$ is initialized according to different structural constraints for each UniCoTT variant:

- **Chain Structure**: $A$ is constrained to represent a linear chain, where each node $i$ is only connected to node $i+1$, forming a sequential path:

$$A_{ij} = \begin{cases} 1 & \text{if } j = i+1 \\ 0 & \text{otherwise} \end{cases} \tag{22}$$

- **Tree Structure**: $A$ is configured as a three-layer binary tree, where each non-leaf node has exactly two children:

$$A_{ij} = \begin{cases} 1 & \text{if } j \text{ is a child of } i \\ 0 & \text{otherwise} \end{cases} \tag{23}$$

- **Graph Structure**: $A$ is initialized as a directed connected graph, where each node $i$ is randomly connected to $k_i$ other nodes ($1 \leq k_i \leq K$), subject to the constraint that the resulting graph remains connected:

$$A_{ij} = \begin{cases} 1 & \text{if } j \text{ is randomly selected as adjacent to } i \\ 0 & \text{otherwise} \end{cases} \tag{24}$$

To ensure connectivity in the graph structure, we employ a modified depth-first search algorithm to verify that all nodes are reachable from the initial node. If the connectivity constraint is not satisfied, the random assignment process is repeated until a valid connected graph is obtained.

---

**Algorithm 1** Algorithm for Iteratively Constructing UniCoT

---

**Input:** The training set $\mathcal{D} = \{< q_i, a_i^* >\}_{i=1}^n$.
**Parameter:** prompt $p_0$ for $v_0$, prompt $p$ for next node, LLMs $p_{LLM}(\cdot)$.
**Output:** UniCoT for each question-answer pair.
  1: **for** each batch $< q, a^* >$ in $\mathcal{D}$ **do**
  2:     Initialize adjacency matrix $\mathcal{A} = \{a_{ij}\}$
  3:     Obtain $v_0 \sim p_{LLM}(v_1|p_0, q, a^*)$
  4:     **for** each non-zero node $v_i$ in $\mathcal{A}$ **do**
  5:         Construct prefix explanations $\mathcal{P}(v_i)$,
  6:         Obtain $v_i \sim p_{LLM}(v_1|p, q, a^*, \mathcal{P}(v_i))$
  7:     **end for**
  8:     Obtain Relation Matrix $\mathcal{R}$ via Eq. 4.
  9: **end for**
 10: **return** Explanations $\mathcal{V} = \{v_i\}$ and Matrix $\mathcal{R}$

---

Table 7: A performance comparison on the factual reasoning benchmark, with the best results emphasized in **bold**. We use XLNet-large as the base model to conduct this experiment.

| Base Model | Method | Structure | CREAK | | | CSQA2 | | | StrategyQA | | |
|---|---|---|---|---|---|---|---|---|---|---|---|
| | | | Acc. | F1 | Ins. | Acc. | F1 | Ins. | Acc. | F1 | Ins. |
| XLNet-large | +None | - | 75.7 | 75.8 | 76.0 | 58.2 | 58.3 | 59.4 | 86.7 | 86.9 | 88.2 |
| | CoT | Chain | 87.8 | 87.8 | 88.2 | 74.9 | 74.8 | 76.1 | 88.5 | 88.5 | 89.4 |
| | SCOTT | Chain | 91.5 | 91.5 | 90.8 | 84.4 | 84.4 | 85.1 | 89.3 | 89.2 | 89.8 |
| | UniCoTT | Chain | 93.4 | 93.4 | 93.7 | 84.6 | 84.7 | 85.3 | 91.2 | 91.1 | 94.3 |
| | UniCoTT | Tree | 94.9 | 94.9 | 95.1 | **89.7** | **89.0** | **91.7** | 94.0 | 94.0 | 95.1 |
| | UniCoTT | Graph | **97.1** | **97.0** | **96.3** | 85.4 | 85.4 | 87.2 | **93.5** | **93.4** | **93.8** |

Table 8: A performance comparison on the Multiple-Choice QA benchmark, with the best results emphasized in **bold**. We use XLNet-large as the base model to conduct this experiment.

| Base Model | Method | Structure | CSQA | | | OBQA | | | QASC | | |
|---|---|---|---|---|---|---|---|---|---|---|---|
| | | | Acc. | F1 | Ins. | Acc. | F1 | Ins. | Acc. | F1 | Ins. |
| XLNet-large | +None | - | 86.4 | 75.6 | 68.8 | 75.0 | 58.3 | 46.6 | 88.1 | 60.3 | 33.3 |
| | CoT | Chain | 88.2 | 80.1 | 75.6 | 82.3 | 75.5 | 71.7 | 92.3 | 76.7 | 61.4 |
| | SCOTT | Chain | 90.9 | 82.7 | 80.8 | 83.5 | 77.2 | 78.2 | 88.4 | 64.6 | 35.1 |
| | UniCoTT | Chain | 91.1 | 85.4 | 84.9 | 82.6 | 76.5 | 77.3 | 93.2 | 84.6 | 79.4 |
| | UniCoTT | Tree | 92.2 | 87.5 | 88.0 | 85.1 | 82.7 | 81.4 | **94** | **84.7** | **80.9** |
| | UniCoTT | Graph | **93.1** | **90.3** | **89.4** | **87.2** | **84.8** | **83.5** | 92.2 | 84.1 | 75.2 |

Code 1: Pseudocode for Iteratively Constructing UniCoT

```
1   def algorithm_for_constructing_unicot(data_loader, language_model):
2       explanations = []
3       relation_matrices = []
4
5       for batch in data_loader:
6           q, a_star = batch
7           adjacency_matrix = initialize_adjacency_matrix()
8
9           # Obtain v0
10          v0 = language_model.forward(q, a_star, p0) # p0 is the initial
                 prompt
11          # v0 holds the explanations
12          explanations.append(v0)
13
14          non_zero_nodes = adjacency_matrix.nonzero()
15          for vi in non_zero_nodes:
16              # Construct prefix explanations P(vi)
17              # Related to the preceding node explanation v_{i-1}
18              prefix_explanations = construct_prefix_explanations(vi)
19
20              # Call LLM s to obtain vi , where p is the prompt for next
                     node
21              vi = language_model.forward(q, a_star, p,
                     prefix_explanations)
22              explanations.left_append(vi)
23
24          # Obtain Relation Matrix R
25          R = compute_relation_matrix(adjacency_matrix)
26          relation_matrices.append(R)
27
28      return explanations, relation_matrices
```

Table 9: A performance comparison on the GLUE benchmark, with the best results emphasized in **bold**. We use XLNet-large as the base model to conduct this experiment.

| Base Model | Methods | Structure | CoLA | RTE | WNLI | MRPC | Average. |
|---|---|---|---|---|---|---|---|
| | +None | - | 58.5 | 69.7 | 53.7 | 82.7 | 66.2 |
| | CoT | Chain | 71.1 | 79.2 | 78.5 | 68.8 | 74.4 |
| | SCOTT | Chain | 79.4 | 90.4 | 91.0 | 91.9 | 88.2 |
| XLNet-large | UniCoTT | Chain | 85.6 | 91.1 | 91.5 | 92.7 | 90.2 |
| | UniCoTT | Tree | 87.8 | 93.5 | **94.6** | **95.0** | 92.7 |
| | UniCoTT | Graph | **92.7** | **94.8** | 94.0 | 93.8 | **93.8** |

## A.4 MORE ABLATION EXPERIMENTS AND ANALYSIS

**Evaluate on different base models.** To evaluate the effectiveness of our method under different model architectures, we use XLNet-large as the student model and then evaluate the knowledge transfer effect of our UniCoTT using the same settings as BERT-base and RoBERTa-base. As shown in Table 7, 8, and 9, we can observe that our method can still achieve significant performance gains under different architectures. Meanwhile, we note that using XLNet as a student model can still achieve consistent performance gains in common sense question answering and oral comprehension tasks.

## A.5 ZERO-SHOT PERFORMANCE ON LLMS

We tested the performance of LLM (gpt-3.5-turbo-1106) on testing and training set of multiple datasets from our experiments using API from Openai, and here are the results as Table 10. Due to the inability to perform training, we employed a zero-shot approach for direct inference. Consequently,

we observed that GPT-3.5 generally underperforms compared to distilled smaller models in most tasks in Section 4.2. However, the results also demonstrate the powerful zero-shot generalization capabilities of LLMs, with performance on multiple question-answering tasks surpassing that on single question-answering tasks.

Table 10: Performance of LLM (gpt-3.5-turbo-1106) on testing and training set of multiple datasets from our experiments using API from openai. We employed a zero-shot approach for direct inference.

|  | Dev | | Train | |
|---|---|---|---|---|
|  | Acc. | F1 | Acc. | F1 |
| CSQA | 69.97 | 69.95 | 69.59 | 69.57 |
| CSQA2 | 61.13 | 59.04 | 64.37 | 62.95 |
| OBQA | 73.80 | 73.82 | 72.26 | 72.22 |
| CREAK | 73.43 | 69.02 | 71.28 | 67.11 |
| StrategyQA | 45.21 | 43.07 | 42.85 | 40.68 |
| QASC | 77.56 | 77.62 | 53.69 | 53.72 |

### A.6 USE DIFFERENT TEACHER MODELS

We use GPT-neox-20B as the teacher model to implement our UniCoTT. We adopt chain-based reasoning to construct UniCoTT and conduct experiments, with results shown in Table 11 and 12. Compared to the baseline models in the paper, our method still achieves significant performance gains. This indicates that our UniCoTT has good generalizability across different teacher models.

Table 11: A performance comparison of the GPT-neox-20B teacher model on the Multiple-Choice QA benchmark . We use BERT-base, RoBERTa-base as the student model to conduct this experiment.

| Base Model | CSQA | | | OBQA | | | QASC | | |
|---|---|---|---|---|---|---|---|---|---|
|  | Acc | F1 | Ins. | Acc | F1 | Ins. | Acc | F1 | Ins. |
| BERT-base | 84.80 | 73.07 | 64.08 | 77.40 | 66.19 | 60.00 | 89.90 | 72.35 | 54.33 |
| RoBERTa-base | 85.55 | 77.99 | 71.58 | 82.75 | 73.80 | 69.20 | 90.60 | 74.30 | 62.11 |

Table 12: A performance comparison of the GPT-neox-20B teacher model on the factual reasoning QA benchmark. We use BERT-base, RoBERTa-base as the student model to conduct this experiment.

| Base Model | CREAK | | | CSQA2 | | | StrategyQA | | |
|---|---|---|---|---|---|---|---|---|---|
|  | Acc. | F1 | Ins. | Acc. | F1 | Ins. | Acc. | F1 | Ins. |
| BERT-base | 78.37 | 78.30 | 78.59 | 71.56 | 71.56 | 71.88 | 83.00 | 82.98 | 82.73 |
| RoBERTa-base | 80.07 | 80.00 | 80.22 | 72.94 | 72.92 | 73.04 | 79.55 | 79.30 | 82.18 |

### A.7 OVERHEAD AND OPERATIONAL EFFICIENCY

We compared the training and inference time with CoT without any method and got the following table. The results in Table 13 are ratios of the training and reasoning times compared to the baseline models. It can be seen that our method introduces only a minimal performance overhead. We believe that the increase in training and inference cost is acceptable compared to the improvement in performance.

### A.8 CASE STUDY OF PROMPTS AND PREDICTIONS

We provide structured explanations and predicted results of our method on CSQA and OBQA datasets in Table 16, Table 17, and Table 18, respectively. The results in the table intuitively demonstrate the efficient transfer ability of structured UniCoTT to superstitious reasoning knowledge, which can

Table 13: Demonstration of the additional time overhead of UniCoTT for training and inference on different architectures compared to CoT.

|  | Chain | Tree | Graph |
|---|---|---|---|
| Training | ×1.21 | ×1.49 | ×1.56 |
| Inference | ×1.15 | ×1.33 | ×1.50 |

greatly improve the performance of SLMs in open-domain question-answering tasks and also have stronger reasoning ability for complex common sense reasoning tasks.

### A.9 MORE DETAILS ON EXPERIMENTING WITH DECODER-ONLY ARCHITECTURE

We utilized the LLaMA-factory [2] framework to implement and train our method. To enable the decoder-only architecture, which follows a next-token prediction paradigm, to use our proposed unified structured CoT, we incorporated our structured CoT as part of the instruction input to the `Qwen2.5-3B-Instruct` model. We then applied Low-Rank Adaptation (LoRA) to perform Supervised Fine-Tuning (SFT) on the model. We formatted the questions (including mathematical problems and multiple-choice questions) as the prefix of the instruction, followed by our generated UniCoT as the instruction suffix. To distinguish between these two parts of the instruction, we employed the special token `[Rationale]`.

In this study, the LoRA configuration was set with a rank of 16 and a dropout rate of 0, targeting the query and value projection matrices of the attention mechanism. This setup allowed for efficient adaptation of the model while maintaining a relatively low number of trainable parameters.

The training process was optimized for computational efficiency and model performance. We utilized a per-device training batch size of 1, combined with gradient accumulation over 8 steps, effectively simulating a larger batch size while conserving memory. The learning rate was set to 1.0e-4, and the model was trained for 3 epochs. To manage the learning rate schedule, we implemented a cosine decay strategy with a warmup ratio of 0.1, allowing for initial rapid learning followed by a gradual decrease in the learning rate. Additionally, we leveraged mixed-precision training using bfloat16 (bf16) to accelerate computations and reduce memory usage without significant loss in model accuracy. The training loss function curves are illustrated in Figure 4.

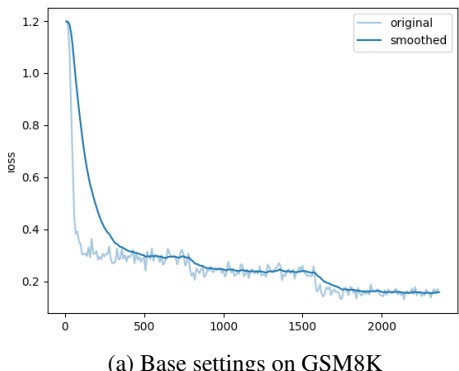
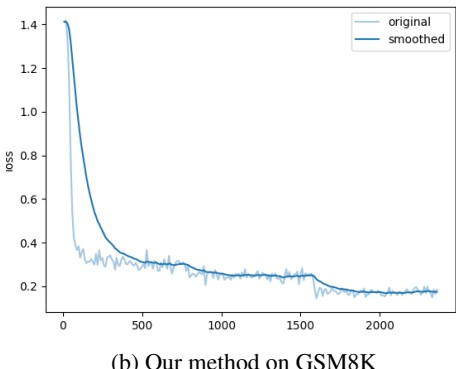

(a) Base settings on GSM8K          (b) Our method on GSM8K

Figure 4: Loss curves obtained by performing SFT training on Qwen2.5-3B-Instruct.

### A.10 STATISTICAL ANALYSIS OF UNICOTT STRUCTURE

To provide quantitative insights into the structural characteristics of UniCoTT, we conducted a comprehensive analysis of the node and edge distributions across different structural variants. Table 14 presents the average number of nodes and edges for chain-structured, tree-structured, and graph-structured UniCoTT configurations.

---

[2] https://github.com/hiyouga/LLaMA-Factory

Table 14: Statistical analysis of nodes and edges in different UniCoTT structures

| Structure | Avg. Nodes | Avg. Edges |
|---|---|---|
| Chain | 4.47 | 3.47 |
| Tree | 7.00 | 6.00 |
| Graph | 8.34 | 10.69 |

For the tree-structured variant, we specifically designed a three-layer binary tree configuration, resulting in a fixed node count of 7. This consistent structure enables systematic comparison across different datasets while maintaining architectural stability. The chain-structured variant exhibits more compact representations with approximately 4-5 nodes on average, while the graph-structured variant demonstrates higher connectivity with an average of 8-9 nodes and 10-11 edges, reflecting its more complex reasoning patterns.

These statistics provide valuable insights into the structural complexity and computational requirements of different UniCoTT variants, helping to inform implementation choices based on specific application constraints and performance requirements.

Table 15: Performance (%) on CREAK dataset with varying number of nodes

| Structure | Number of Nodes | | | | | |
|---|---|---|---|---|---|---|
| | 2 | 3 | 4 | 5 | 6 | 7 |
| Chain | 47.49 | 51.47 | 52.56 | 52.80 | - | - |
| Tree | 47.43 | 51.55 | 53.39 | 54.91 | 56.60 | 56.71 |

## A.11 ANALYSIS OF NODE COUNT (KNOWLEDGE SIZE) IMPACT

To investigate the relationship between structural complexity and model performance, we conducted an ablation study on the CREAK dataset, focusing on chain-structured and tree-structured variants of UniCoTT. These variants were selected for their amenability to systematic node expansion compared to the graph-structured configuration. For computational efficiency, we utilized a 10% random sample of the CREAK dataset.

Table 15 presents the performance trends across different node configurations. For chain-structured UniCoTT, performance stabilizes at approximately 4 nodes, achieving an accuracy of 52.56%. The tree-structured variant demonstrates continuous improvement up to 7 nodes, reaching 56.71% accuracy. Notably, the performance gains begin to plateau after reaching these respective node counts, suggesting diminishing returns with further structural expansion.

These empirical findings indicate that effective knowledge distillation can be achieved with moderate structural complexity: 4 nodes for chain structures and 7 nodes for tree structures represent optimal configurations that balance performance gains with computational efficiency. This observation has important implications for practical implementations, as it suggests that the benefits of structural reasoning can be realized without requiring excessive computational resources or annotation effort.

| | |
|---|---|
| **Question** | What does a gambler do that causes him or her to be unhappy? (A) play cards (B) double winnings (C) lose money (D) play poker (E) to win the prize |
| **Gold Answer** | C |
| **Structure of UniCoT** |  |
| **Prompt for $v_0$** | Q: what will move to another area if their habitat will no longer support them?\nAnswer choices:\n(a) density\n(b) Birds\n(c) squids\n(d) humans\n(e) clouds\n(f) gravity\n(g) cows\n(h) Whales\nA:The answer is cows\nThe explanation is:If a habitat can no longer support animals then those animals will move to another area.\n\n
Q: Question \n A: The answer is C. The brief explanation is: |
| **Prompt for next node** | Q: what will move to another area if their habitat will no longer support them?\nAnswer choices:\n(a) density\n(b) Birds\n(c) squids\n(d) humans\n(e) clouds\n(f) gravity\n(g) cows\n(h) Whales\nA:The answer is cows\nThe brief explanation is:If a habitat can no longer support animals then those animals will move to another area.\nA brief explanation of another perspective is:Cows are social animals.\n
Q: Question \n A: The answer is C. The brief explanation is: Prompt for v_0 Parent node explanation \nA brief explanation of further perspective is: |
| $v_0$ | Gamblers may feel unhappy when losing money because it goes against their desire to win. |
| $v_1$ | The answer is lose money. A gambler may experience unhappiness when they lose money, as it goes against their desire to win and can lead to financial setbacks |
| $v_2$ | The question seems incomplete, and there is no further perspective provided for the second question. If you have additional information or if there's anything specific you'd like to know or discuss regarding the second question, please provide more details. |
| $v_3$ | The brief explanation provided for the second question is: "The answer is lose money. A gambler may experience unhappiness when they lose money, as it goes against their desire to win and can lead to financial setbacks |
| $v_4$ | The act of losing money can cause a gambler to be unhappy because it means they have not been successful in their attempt to win, and it can also lead to financial difficulties |
| $v_5$ | It seems like you've provided a question about a gambler and its answer regarding losing money. |
| $v_6$ | Losing money can also have a psychological impact on a gambler, causing stress, frustration, and even feelings of failure. Some gamblers may also experience guilt or shame for losing money, especially if it affects their family or financial stability. This emotional toll can contribute to their unhappiness as well. |
| **Prediction of +UniCoTT** | **C** |
| **Prediction of +CoT** | **D** |
| **Prediction of +SCOTT** | **D** |

Table 16: We demonstrate the graph structure of UniCoTT and the explanations of various nodes generated by LLMs on the CSQA dataset. Meanwhile, we present the prediction results of different methods using the BERT-base as the backbone network.

| Question | James needed smooth sandpaper, but instead he got what type? (A) rough (B) non smooth (C) uneven (D) plastic (E) bumpy |
|---|---|
| **Gold Answer** | A |
| **Structure of UniCoT** |  |
| **Prompt for $v_0$** | Q: what will move to another area if their habitat will no longer support them?\nAnswer choices:\n(a) density\n(b) Birds\n(c) squids\n(d) humans\n(e) clouds\n(f) gravity\n(g) cows\n(h) Whales\nA:The answer is cows\nThe explanation is:If a habitat can no longer support animals then those animals will move to another area.\n\n
Q: Question \n A: The answer is A. The brief explanation is: |
| **Prompt for next node** | Q: what will move to another area if their habitat will no longer support them?\nAnswer choices:\n(a) density\n(b) Birds\n(c) squids\n(d) humans\n(e) clouds\n(f) gravity\n(g) cows\n(h) Whales\nA:The answer is cows\nThe brief explanation is:If a habitat can no longer support animals then those animals will move to another area.\nA brief explanation of another perspective is:Cows are social animals.
Q: Question \n A: The answer is A. The brief explanation is: Prompt for v_0 Parent node explanation \nA brief explanation of further perspective is: |
| $v_0$ | James needs smooth sandpaper, but he may get the opposite smoothness |
| $v_1$ | James needed smooth sandpaper, but instead, he got rough sandpaper. |
| $v_2$ | The answer is (a) rough.\n\nThe brief explanation is: James needed smooth sandpaper, but instead, he got rough sandpaper.\n\nA brief explanation of further perspective is: Smooth and rough are contrasting textures, and in this context, James received the opposite of what he needed, indicating rough sandpaper. |
| $v_3$ | The given responses appear to be mixed up. Let me organize and provide concise answers: James needed smooth sandpaper, but instead, he got rough sandpaper. Smooth and rough are contrasting textures, and in this context, James received the opposite of what he needed, indicating rough sandpaper. |
| $v_4$ | James needed smooth sandpaper, but instead, he got rough sandpaper. Smooth and rough are contrasting textures, and in this context, James received the opposite of what he needed, indicating rough sandpaper |
| $v_5$ | It seems like you've provided a question about a gambler and its answer regarding losing money. |
| $v_6$ | The concept of smooth is being contrasted with rough in this question, and the correct answer is rough. Smooth and rough are opposites in terms of texture, and in this case, James received the opposite of what he needed, indicating rough sandpaper. |
| **Prediction of UniCoTT** | A |
| **Prediction of CoT** | **B** |
| **Prediction of SCOTT** | A |

Table 17: We demonstrate the tree structure of UniCoTT and the explanations of various nodes generated by LLMs on the CSQA dataset. Meanwhile, we present the prediction results of different methods using the BERT-base as the backbone network.

| Question | Where would animals shiver the most? (A) Arctic Circle (B) Sumatra (C) Java (D) tropical rainforest |
|---|---|
| **Gold Answer** | A |
| **Structure of UniCoT** | $v_0 \longrightarrow v_1 \longrightarrow v_2 \longrightarrow v_3$ |
| **Prompt for $v_0$** | Q: what will move to another area if their habitat will no longer support them?\nAnswer choices:\n(a) density\n(b) Birds\n(c) squids\n(d) humans\n(e) clouds\n(f) gravity\n(g) cows\n(h) Whales\nA:The answer is cows\nThe explanation is:If a habitat can no longer support animals then those animals will move to another area.\n\n
Q: Question \n A: The answer is A. The brief explanation is: |
| **Prompt for next node** | Q: what will move to another area if their habitat will no longer support them?\nAnswer choices:\n(a) density\n(b) Birds\n(c) squids\n(d) humans\n(e) clouds\n(f) gravity\n(g) cows\n(h) Whales\nA:The answer is cows\nThe brief explanation is:If a habitat can no longer support animals then those animals will move to another area.\nA brief explanation of another perspective is:Cows are social animals.
Q: Question \n A: The answer is A. The brief explanation is: Prompt for v_0 Parent node explanation \nA brief explanation of further perspective is: |
| $v_0$ | If a habitat can no longer support animals then those animals will move to another area. |
| $v_1$ | Animals would shiver the most in the Arctic Circle because of the extremely cold temperatures in that region. |
| $v_2$ | Animals would shiver the most in the Arctic Circle because it is the coldest region listed. |
| $v_3$ | The Arctic Circle is known for its extreme cold temperatures, making it the most likely place for animals to shiver.Animals in the Arctic Circle have adapted to survive in extremely cold conditions, so they would shiver the most in this environment |
| **Prediction of UniCoTT** | A |
| **Prediction of CoT** | D |
| **Prediction of SCOTT** | C |

Table 18: We demonstrate the chain structure of UniCoTT and the explanations of various nodes generated by LLMs on the OBQA dataset. Meanwhile, we present the prediction results of different methods using the BERT-base as the backbone network.

