# OpenReview forum: "UniCoTT: A Unified Framework for Structural Chain-of-Thought Distillation"
_ICLR.cc/2025/Conference — ICLR 2025 Poster_

### Official Review · Reviewer_546m · 2024-10-28

**Soundness:** 3
**Presentation:** 3
**Contribution:** 3
**Rating:** 6
**Confidence:** 2

**Summary:**

This paper  introduces UniCoTT, a teacher-student framework aimed at transferring complex reasoning abilities from large language models (LLMs) to smaller language models (SLMs). UniCoTT extends traditional chain-of-thought (CoT) reasoning by leveraging diverse structured reasoning paths, such as chains, trees, and graphs, within a unified distillation process. This approach involves iterative CoT construction, node-level supervised contrastive learning, and structural consistency learning to reinforce reasoning capabilities in SLMs. Experimental results on factual reasoning, multiple-choice QA, and natural language understanding tasks demonstrate that UniCoTT outperforms existing methods, enhancing SLM performance across several benchmarks.

**Strengths:**

- Extends CoT reasoning with diverse structures, which broadens the reasoning capabilities of SLMs.
- Implements structural consistency and contrastive learning, effectively aligning SLMs with complex CoT reasoning paths.
- Demonstrates superior performance on multiple tasks, showing effectiveness and generality in knowledge transfer.

**Weaknesses:**

UniCoTT’s increased complexity and computational requirements could make real-world deployment challenging. To be fair, as distillation strategy proposed in this paper uses three types of reasoning and more compute to create dense supervision. The baselines like CoT may also uses more compute like more chains in self-consistency to increase the quality of distillation data.

**Questions:**

No

---

> ### Author Response · Authors · 2024-11-20
> **Response to Reviewer 546m**
>
> We sincerely appreciate your valuable feedback and for acknowledging the strengths of our work, including the extension of CoT reasoning with diverse structures, the effective use of structural consistency and contrastive learning, and the superior performance demonstrated across multiple tasks. Your constructive comments provide meaningful insights, and we address each of them in detail below to further improve the clarity and applicability of our proposed approach.
>
> ---
>
> **[W1]** UniCoTT’s increased complexity and computational requirements could make real-world deployment challenging. To be fair, as distillation strategy proposed in this paper uses three types of reasoning and more compute to create dense supervision. The baselines like CoT may also uses more compute like more chains in self-consistency to increase the quality of distillation data.
>
> **[A1]**
> Thank you for this question:
>
> (1) In fact, across all experiments in our paper, the number of explanation nodes generated for CoT distillation and the SCOTT method is consistent with the number of explanation nodes in our chain-like UnCoTT. Nevertheless, our method achieved more effective results. This further demonstrates the efficacy of our generation rationale and the strategy of distilling to smaller models.
>
> (2) Our tree and graph structural UniCoTT indeed introduce more explanation nodes and dense computations compared to chain-like methods. To further investigate the relationship between the introduced computational density and distillation effects, we expanded the chain-like explanation nodes in CoT distillation methods to be almost consistent with the tree and graph structures for experimentation. Specifically, we configured the node count to 7, which entails generating more extensive explanatory chains for chain-like UniCoTs.
> We evaluated their results on the CREAK dataset and obtained 89.32% accuracy, which is still lower than the results of UniCoTT in Table 1 of our paper. It can be observed that despite having the same annotated explanations, our method still demonstrates superior performance.

---

> > ### Comment · Reviewer_546m · 2024-11-23
> >
> > Thanks for the explanation!

---

> > > ### Author Response · Authors · 2024-11-24
> > > **Gratitude for Your Constructive Feedback**
> > >
> > > Dear Reviewer 546m,
> > >
> > > We sincerely appreciate your valuable comments and insightful feedback on our work. Your thoughtful suggestions have significantly contributed to improving the quality of our paper. We are grateful for your time and effort in reviewing our manuscript.
> > >
> > > Wishing you continued success in your endeavors!
> > >
> > > Best regards,
> > > Team of paper #5668

---

### Official Review · Reviewer_LLAY · 2024-11-01

**Soundness:** 3
**Presentation:** 2
**Contribution:** 2
**Rating:** 6
**Confidence:** 4

**Summary:**

This paper proposes UniCoTT, a unified distillation framework to transfer the diverse reasoning structures with CoT to smaller language models such as BERT and RoBERTa. Firstly, UniCoT is proposed as a unified bridge of various CoT structures, which is constructed by iteratively prompting LLMs to produce explanations with correct answers. After that, a node-level supervised contrastive loss and a structural consistency loss are designed as part of the training objective. Experiments on multiple reasoning datasets verified the effectiveness of UniCoTT by surpassing the baseline methods by a large margin.

**Strengths:**

1. The proposed method is technically sound and intuitively makes sense. It is very interesting to transfer the knowledge from structured CoT texts into smaller models that can leverage rationale knowledge in a unified manner.
2. Experimental results on several benchmarks show some improvement upon baselines and the authors conduct extensive ablation studies and analyses on various design choices.
3. The paper itself is generally well-written.

**Weaknesses:**

1. The generalizability of KNIFE is yet to be known. The proposed framework is only verified in multiple-choice datasets. Whether it could be extended to other task settings like text generation remains a concern.
2. The process of iteratively constructing UniCoT is hard to understand from the main body of the current version. I would suggest the authors move some content from the appendix to the main body. Meanwhile, it would be helpful if the authors could provide some overall statistics on the constructed UniCoT. For example, the averaged nodes and edges of the structure.

**Questions:**

1. The authors list "hallucinations" as one of the major drawbacks of previous works, and motivate the design of UniCoTT in ``introduction'' section. I am wondering how the designed UniCoTT framework helps to alleviate this issue.
2. In lines 385-386, why $\alpha$ and $\beta$ is set to 0.5 and 0.2 respectively? Is it an intuitive trial or a result of a grid search?
3. It would be interesting to test the annotation efficiency of CoT with the teacher model. An empirical conclusion of how many annotations are enough for great distillation performance would be insightful.

---

> ### Author Response · Authors · 2024-11-20
> **Response to Reviewer LLAY [1/3]**
>
> We sincerely appreciate your detailed feedback and for highlighting the strengths of our work, including the technical soundness of our proposed method, the interest in transferring structured CoT knowledge into smaller models, and the extensive ablation studies conducted. Your constructive comments and questions are highly valuable, and we address each of them below to further clarify our contributions and improve the manuscript.
>
> ---
>
> **[W1]** The generalizability of KNIFE is yet to be known. The proposed framework is only verified in multiple-choice datasets. Whether it could be extended to other task settings like text generation remains a concern.
>
> **[A1]**
> While our UniCoTT framework was initially designed and validated for classification tasks, including factual reasoning, multiple-choice QA, and NLU tasks, we have conducted additional experiments to evaluate its generalizability to text generation scenarios.
>
> To comprehensively assess the extensibility of our approach, we evaluated various tasks including mathematical reasoning, commonsense reasoning, and open-domain question answering using a text generation paradigm. We employed Qwen2.5-3B-Instruct as our foundation model, utilizing our generated graph-structured UniCoT as instruction input and incorporating structural constrained adjacency matrices as additional prompts into the decoder model architecture. The model training adhered to the next-token prediction paradigm through supervised fine-tuning (SFT) with low-rank adaptation (LoRA), implemented using LLaMA factory.
>
> Our experimental results demonstrate consistent improvements across various tasks, including factual reasoning and GSM8K mathematical reasoning benchmarks, compared to the base model. These improvements consistently validate the effectiveness of our method across diverse task domains. For comprehensive results, analysis, experimental details, and training loss curves, we refer you to our revised manuscript. To facilitate result reproduction, we provided corresponding configurations and code. Additional experimental configurations and implementation details are included in the supplementary materials.
>
> | Model | Factual Reasoning | Multi-choice QA | | | | Mathematical Reasoning |
> |-------|-------------------|-----------------|-----------------|-------|-------|---------------------|
> | | CREAK | CSQA2 | StrategyQA | CSQA | OBQA | GSM8K |
> | Base | 88.8 | 63.7 | 83.2 | 92.0 | 91.0 | 76.9 |
> | UniCoTT (Ours) | 91.5 | 75.4 | 88.7 | 95.0 | 92.9 | 79.2 |
>
> ---
>
> **[W2]** The process of iteratively constructing UniCoT is hard to understand from the main body of the current version. I would suggest the authors move some content from the appendix to the main body. Meanwhile, it would be helpful if the authors could provide some overall statistics on the constructed UniCoT. For example, the averaged nodes and edges of the structure.
>
> **[A2]**
>
> (1) Thank you for the professional opinion. We have added more detailed information on the construction method of UniCoT in section 3.2. However, due to the page limit of the main text, we will still keep the algorithm pseudocode and other content in the appendix.
>
> (2) We appreciate the reviewer's constructive suggestion regarding the statistical characteristics of our UniCoTT structure. To address this concern and provide a more comprehensive analysis, we have conducted a thorough examination of the node and edge counts in our generated UniCoTT structures across all datasets. This analysis is particularly valuable given that the length of explanations generated by Large Language Models (LLMs) can vary, and fewer invocations may sometimes produce longer explanations than multiple calls. We conducted a statistical analysis of the number of nodes and edges in the UniCoTT structure we generated on all the datasets, shown below the table:
>
> | Structure | Nodes | Edges |
> |-----------|-------|-------|
> | Chain | 4.47 | 3.47 |
> | Tree | 7.00 | 6.00 |
> | Graph | 8.34 | 10.69 |
>
> We have saved this detail in the revised version. It is worth noting that for the tree structure, the node count is consistently 7. This is due to our experimental design, where we constrained the tree-structured chain of thought to a three-layer binary tree.
>
> ---

---

> ### Author Response · Authors · 2024-11-20
> **Response to Reviewer LLAY [2/3]**
>
> **[Q1]** The authors list "hallucinations" as one of the major drawbacks of previous works, and motivate the design of UniCoTT in ``introduction'' section. I am wondering how the designed UniCoTT framework helps to alleviate this issue.
>
> **[A3]**
> Thank you for your insightful comment. We acknowledge the potential for hallucination in Large Language Models (LLMs), which can lead to inaccuracies in generated content. If the explanations generated by LLMs do not accurately align with factual information, they may fail to provide positive training signals for smaller models and could potentially degrade their performance due to error propagation. Therefore, maintaining the rationality and fidelity of LLM outputs during the distillation process is crucial.
> Motivated by this concern, we have implemented methods to ensure the reasonableness of LLM-generated explanations, as detailed in Section 3.2 of our manuscript:
> 1. Following the SCOTT approach, we utilize annotated question-answer pairs $<p, q, a*>$ as prompts for LLM explanation generation.
> 2. We guide the LLM to adhere to a structured reasoning process when generating explanations, ensuring that the relationships between explanations are more accurately represented by the adjacent order matrix.
> These strategies have enabled our method to generate more rational explanations, which further elucidates why our chain-like structure outperforms vanilla CoT distillation methods.
> Furthermore, in Section 4.3 of our manuscript, we present quantitative experiments evaluating the rationality of explanations constructed by our proposed method, as shown in Table 5. The results demonstrate that our construction method produces explanations with lower hallucination rates and higher fidelity.This comprehensive approach not only addresses the potential limitations of LLM-generated content but also provides empirical evidence for the effectiveness of our method in maintaining explanation quality throughout the distillation process.
>
> ---
>
> **[Q2]** Why $\alpha$ and $\beta$ is set to 0.5 and 0.2 respectively? Is it an intuitive trial or a result of a grid search?
>
> **[A4]**
> Thank you for your insightful question. The parameters $\alpha$ and $\beta$ govern the balance between supervised learning (including supervised contrastive learning and cross-entropy) and structural constraints. Our hyperparameter selection was conducted through a systematic grid search within a predetermined range during the experimental process. Specifically, we first performed a grid search for $\alpha$ within the range [0.1, 0.9] on the CREAK dataset. After determining the relatively optimal value of $\alpha=0.5$, we then conducted a grid search for $\beta$ within the same range [0.1, 0.9] and obtain the optimal $\beta=0.2$. Subsequently, we applied these optimal parameters derived from the CREAK dataset to other datasets in our study. We acknowledge that executing individual grid searches for each dataset could potentially yield even more favorable results.
> We have incorporated these methodological details into our manuscript to provide a more comprehensive account of our hyperparameter tuning process.
>
> ---
>
> **[Q3]** It would be interesting to test the annotation efficiency of CoT with the teacher model.
>
> **[A5]**
> The annotation efficiency of our approach, which primarily utilizes ChatGPT-3.5-turbo (and GPT-Neo-20B for some experiments) as the teacher model, is determined by two key factors:
> a) Single API call latency;
> b) Total number of API calls required for generating explanations.
>
> Regarding (a), the throughput is primarily constrained by network bandwidth and API service capacity. In our experimental setup using a server with gigabit network connectivity, we achieved an average response time of approximately 2.1s per API call (with a token limit of 512). Using parallel processing with 5 concurrent threads, we can complete the annotation of 1,000 structured UniCoT samples with 7 nodes each in approximately 52 minutes.
>
> For (b), we analyzed the average number of nodes and edges in our UniCoT structures, as shown in **[A2]** The number of nodes represents the API calls required to generate explanations for each sample. Our observations indicate that the API call requirements remain relatively modest, ensuring efficient construction of UniCoTT.

---

> ### Author Response · Authors · 2024-11-20
> **Response to Reviewer LLAY [3/3]**
>
> ---
> **[Q4]** An empirical conclusion of how many annotations are enough for great distillation performance would be insightful.
>
> **[A6]**
> To investigate the relationship between node count and distillation performance, we conducted experiments using chain-structured and tree-structured UniCoTT (which are more amenable to node expansion compared to graph-structured UniCoTT) on the CREAK dataset. For computational efficiency, we sampled 10% of the CREAK dataset. Our empirical results, as shown below, demonstrate that optimal performance can be achieved with relatively modest node counts. Specifically, 4 nodes for chain structures and 7 nodes for tree structures yield optimal performance while maintaining reasonable computational efficiency.
>
> | Structure/Nodes | 2 | 3 | 4 | 5 | 6 | 7 |
> |----------------|------|------|------|------|------|------|
> | Chain | 47.49 | 51.47 | 52.56 | 52.80 | - | - |
> | Tree | 47.43 | 51.55 | 53.39 | 54.91 | 56.60 | 56.71 |
>
>
> These results indicate that effective knowledge distillation can be achieved with a moderate number of explanation nodes, establishing an optimal balance between performance and annotation efficiency.

---

> ### Author Response · Authors · 2024-11-25
> **Thank you for your efforts.**
>
> Dear Reviewer LLAY,
>
> We are truly appreciative of the time and effort you have dedicated to reviewing our paper. Your thoughtful feedback and constructive suggestions are valuable to us. We have carefully addressed your comments in our rebuttal to enhance the quality of our work. As we approach the final days of the Author-Review Discussion period, we would like to ensure that all your concerns have been comprehensively addressed. Should there be any remaining questions or issues, we are willing to provide further clarification or additional revisions.Thank you once again for your insightful contributions to our work.
>
> Best regards,
>
> Team 5668

---

> ### Comment · Reviewer_LLAY · 2024-11-25
>
> Thanks for the detailed explanation and the experiments. I think most of my concerns are addressed. I appreciate the authors' effort in sharing their thorough analysis and valuable insights. I think my evaluation is fair and I changed my confidence score.

---

### Official Review · Reviewer_wQ9A · 2024-11-03

**Soundness:** 3
**Presentation:** 2
**Contribution:** 2
**Rating:** 5
**Confidence:** 4

**Summary:**

The paper presents a novel framework for transferring the reasoning capabilities of large language models (LLMs) to small language models (SLMs) through a structured chain-of-thought (CoT) distillation approach. The authors propose UniCoTT, which considers diverse structural CoTs (chain, tree, and graph) and employs two core strategies: iterative construction for structured CoTs and a structural constraint strategy. The framework aims to address the challenges of ensuring the rationality of generated explanations and ignoring diverse structures of CoT during knowledge transfer. The experimental results demonstrate significant performance improvements of SLMs on multiple NLP tasks across various datasets.

**Strengths:**

1. The paper introduces a unified framework that handles diverse structural CoTs, which is a significant advancement over existing methods that focus solely on chain structures.
2. The authors provide extensive experimental evidence to support the effectiveness of UniCoTT, showing improvements across different NLP tasks and datasets.
3. The consideration of structured reasoning pathways (tree and graph) in addition to chains is a strength, as it better captures the complexity of human reasoning processes.

**Weaknesses:**

1. The paper could benefit from a discussion on the computational complexity of UniCoTT and its scalability, especially when dealing with very large datasets or more complex reasoning tasks.
2. The construction of UniCoT relies on APIs of LLMs, which may not be accessible or feasible in all situations. The paper could address potential alternatives or mitigation strategies. Besides, SLMs usually refer to small language models, e.g., 2B and 3B. The authors mainly conducted experiments on BERT and RoBERTa, which were not convincing enough.
3. While the results are promising, the paper primarily focuses on question-answering and NLU tasks. It would be beneficial to see how UniCoTT generalizes to other types of tasks.

**Questions:**

1. How does the performance of UniCoTT scale with the size and complexity of the knowledge to be transferred? Are there diminishing returns as the complexity increases?
2. What are the limitations of the current implementation of UniCoTT, and how might these be addressed in future work?

---

> ### Author Response · Authors · 2024-11-20
> **Response to Reviewer wQ9A [1/2]**
>
> We sincerely appreciate your detailed and constructive feedback. We are grateful that you recognized the strengths of our work, including the introduction of a unified framework that handles diverse structural CoTs, the extensive experimental evidence demonstrating UniCoTT's effectiveness across different NLP tasks, and our consideration of structured reasoning pathways (tree and graph) that better reflect human reasoning. Your comments and questions are highly valuable, and we address them thoroughly in the responses below to enhance the clarity and completeness of our work.
>
> ---
>
> **[W1]**  The paper could benefit from a discussion on the computational complexity of UniCoTT and its scalability, especially when dealing with very large datasets or more complex reasoning tasks.
>
> **[A1]**
> Thank you for this valuable suggestion. We would like to address both the computational complexity and scalability aspects of our method:
>
> (1) Regarding computational complexity during training, as analyzed in Appendix A7, our method introduces only marginal overhead compared to distillation without CoT. Specifically, training small models using chain-structured, tree-structured, and graph-structured UniCoTT incurs computational costs of 1.21x, 1.49x, and 1.56x respectively, compared to distillation without CoT. Therefore, while our method does introduce additional computational overhead, it remains acceptable given the significant performance gains achieved.
>
> (2) Concerning scalability, we conducted additional experiments using Qwen2.5-3b-Instruct for question-answering and mathematical reasoning tasks. On the GSM8K dataset, our method demonstrated superior performance compared to approaches without UniCoTT. This further validates that our method maintains strong performance across more complex tasks and different architectures. Please refer to response **[A2]** and our revised manuscript for more detailed analysis and results.
>
> ---
>
> **[W2]** The construction of UniCoT relies on APIs of LLMs, which may not be accessible or feasible in all situations. The paper could address potential alternatives or mitigation strategies. Besides, SLMs usually refer to small language models, e.g., 2B and 3B.
>
> **[A2]**
> (1) As discussed in our limitations section, while our primary experiments utilize OpenAI's API, we acknowledge this dependency might not be universally accessible. To address this concern,  in the original manuscript,  we have conducted additional experiments using the open-source GPT-NeoX-20B as the teacher model, as detailed in Appendix A6 (Tables 10 and 11). Comparing these results with Tables 1 and 2 in the main manuscript, our method consistently outperforms conventional CoT distillation and SCOTT approaches, demonstrating its effectiveness even with less powerful, open-source LLMs.
>
> (2) We appreciate this professional inquiry. To further evaluate the efficacy of our approach, we employed the decoder-only Qwen2.5-3B-Instruct as our foundation model. Specifically, we utilized our generated graph-based UniCoT as instruction input for Qwen2.5-3B-Instruct and incorporated our structural constrained adjacency matrix as additional prompts into the decoder-only model architecture. The model training adhered to the next-token prediction paradigm via supervised fine-tuning (SFT) with low-rank adaptation (LoRA), implemented using LLaMA factory.
> As shown in the table below, our experimental results demonstrate consistent improvements across various tasks, including factual reasoning, multi-choice QA, and mathematical reasoning (i.e., GSM8K)  benchmarks, compared to the base model. These improvements validate the effectiveness of our method across diverse tasks. For detailed results, analysis, experimental configurations, and training loss curves, please refer to our revised manuscript and supplementary materials.
>
> | Model | Factual Reasoning | Multi-choice QA | | | | Mathematical Reasoning |
> |-------|-------------------|-----------------|-----------------|-------|-------|---------------------|
> | | CREAK | CSQA2 | StrategyQA | CSQA | OBQA | GSM8K |
> | Base | 88.8 | 63.7 | 83.2 | 92.0 | 91.0 | 76.9 |
> | UniCoTT (Ours) | 91.5 | 75.4 | 88.7 | 95.0 | 92.9 | 79.2 |
>
> ---
>
> **[W3]**  It would be beneficial to see how UniCoTT generalizes to other types of tasks.
>
> **[A3]**
> Beyond question-answering and NLU tasks, we have extended our evaluation to mathematical reasoning tasks. As detailed in response **[A2]**, our method demonstrates significant performance improvements on the GSM8K mathematical reasoning benchmark. This empirical evidence further validates the generalizability of our UniCoT strategy across diverse task domains. Consistent performance gains across these fundamentally different tasks - from natural language understanding to structured mathematical reasoning - substantiate the robustness and transferability of our approach.

---

> > ### Comment · Reviewer_wQ9A · 2024-11-24
> > **Response**
> >
> > Thank you for your reply. I stand by my perspectives and am deeply concerned about the issues I have raised.

---

> > > ### Author Response · Authors · 2024-11-24
> > > **Follow-up on Addressing Your Feedback with Clarifications**
> > >
> > > Dear Reviewer wQ9A,
> > >
> > > We sincerely appreciate the time and effort you dedicated to reviewing our paper and providing thoughtful feedback. Your detailed comments have been invaluable, and we have carefully addressed each of your points in our rebuttal.
> > >
> > > If there are any remaining concerns or areas where you feel we have not fully resolved your feedback, we would greatly appreciate it if you could specify them. We are more than willing to provide further clarifications or make additional revisions to address your concerns.
> > >
> > > Thank you again for your insightful contributions.
> > >
> > > Best regards,
> > >
> > > Team of paper #5668

---

> > > ### Author Response · Authors · 2024-11-27
> > > **Sincerely Hoping for Your Response Regarding Our Rebuttal**
> > >
> > > Dear Reviewer wQ9A,
> > >
> > > We sincerely thank you for the time and effort you have dedicated to reviewing our paper. We have carefully addressed your concerns in our rebuttal to improve the quality of our work.
> > >
> > > In our response, we have specifically addressed your concerns as follows:
> > > - **For Weakness 1**: Discussed and implemented our method on other types of reasoning tasks, including more complex scenarios such as mathematical reasoning.
> > > - **For Weakness 2**: Conducted experiments based on QWen2.5-3B-Instruct as the foundational model.
> > > - **For Weakness 3**: Clarified our experiments on alternatives to using LLM APIs (as previously discussed in the supplementary materials of the initial manuscript).
> > >
> > > Additionally, we have responded to the issues you raised, including:
> > > - **For Question 1**: The performance of UniCoTT with varying scales and complexities of knowledge.
> > > - **For Question 2**: The current limitations of UniCoTT, as discussed in detail.
> > >
> > > As the discussion period for author comments approaches its final days, we want to ensure that all your concerns have been fully addressed. If you have any further questions or require additional clarification, we are more than willing to provide further explanations or revisions.
> > >
> > > Once again, thank you for your profound contributions to improving our work.
> > >
> > > Best regards,
> > >
> > > Team 5668

---

> ### Author Response · Authors · 2024-11-20
> **Response to Reviewer wQ9A [2/2]**
>
> **[Q1]**  How does the performance of UniCoTT scale with the size and complexity of the knowledge to be transferred? Are there diminishing returns as the complexity increases?
>
> **[A4]**
> Thank you for the professional review. We would like to address your question as follows:
>
> (1) We measure the size of transferred knowledge by the number of explanation nodes in our constructed UniCoT. To investigate this relationship, we conducted experiments using chain-structured and tree-structured UniCoTT (which are more amenable to node expansion compared to graph-structured UniCoTT) on the CREAK dataset. For computational efficiency, we sampled 10% of the CREAK dataset to examine the relationship between node count and distillation performance. As shown in the table below, our empirical observations reveal that performance gains initially increase with the number of explanation nodes but eventually plateau, suggesting a diminishing returns effect.
>
> (2) While quantifying knowledge complexity remains challenging, our observations indicate that performance improvements are generally more modest for more complex tasks compared to simpler ones. For instance, as shown in Tables 1 and 2 of our manuscript, the positive performance gains achieved by our method on multiple-choice QA tasks are smaller than those observed for factual reasoning (binary inference) tasks. This pattern suggests that knowledge distillation may be more challenging for complex tasks with intricate explanations or knowledge structures, resulting in less pronounced improvements compared to simpler tasks with more straightforward knowledge transfer requirements.
>
> ---
>
> **[Q2]** What are the limitations of the current implementation of UniCoTT, and how might these be addressed in future work?
>
> **[A5]**
> As stated in the limitations section of our manuscript, the construction of UniCoT relies on APIs of LLMs, which may not be easy to implement in specific situations. Therefore, this article's future research direction is exploring more efficient and low-resource methods. This article studies factual reasoning, open-domain multiple-choice question answering, natural language understanding, and mathematical reasoning tasks. Further research can be conducted in more fields (e.g., code generation and completion).
> Moreover, we have also realized that the structural constraint loss we proposed for classification tasks faces some difficulties when transferred to generative paradigms. This could potentially be addressed by adding extra constraints during the process of predicting the next token or by designing preference learning strategies adapted to structured CoT. As such, designing optimization constraints suitable for small decoder-only models is also one of our future directions.

---

### Official Review · Reviewer_AdyG · 2024-11-04

**Soundness:** 3
**Presentation:** 3
**Contribution:** 3
**Rating:** 8
**Confidence:** 4

**Summary:**

This paper focuses on distilling the reasoning capability, specifically chain-of-thought reasoning, from large language models into smaller models. Specifically, the paper uses prompts to guide a larger teacher model to generate multiple explanations, or "thoughts," for given questions and answers. These explanations are represented in a graph structure. Then, the small student model is trained using traditional cross-entropy loss along with a novel structural consistency loss and supervised contrastive loss proposed by the authors.

**Strengths:**

- The writing is clear and easy to follow, with a well-defined motivation for the research.
- The distillation framework proposed is innovative, especially in using a graph structure to represent different chains of thought and introducing corresponding training methods.
- The approach is extensively tested on multiple benchmark datasets, demonstrating strong empirical performance.

**Weaknesses:**

- The framework mainly focuses on distilling explanation and reasoning abilities into base models like BERT. A concern is the limited application scope of such encoder-based models. To further validate the effectiveness of the proposed distillation framework for reasoning abilities, it would be interesting to distill the chain-of-thought reasoning from larger models into smaller decoder-based models and test them on complex reasoning tasks.

**Questions:**

- Why focus on using an encoder as the student model?

---

> ### Author Response · Authors · 2024-11-20
> **Response to Reviewer AdyG**
>
> We sincerely thank you for your constructive feedback and for highlighting the strengths of our work, including the "clear and easy to follow" writing, the "innovative" distillation framework utilizing a graph structure, and the strong empirical performance demonstrated across multiple benchmark datasets. Your suggestions and questions are valuable, and we address them below to further clarify and enhance our contributions.
>
> ---
> **[W1]** The framework mainly focuses on distilling explanation and reasoning abilities into base models like BERT. A concern is the limited application scope of such encoder-based models. To further validate the effectiveness of the proposed distillation framework for reasoning abilities, it would be interesting to distill the chain-of-thought reasoning from larger models into smaller decoder-based models and test them on complex reasoning tasks.
>
> **[A1]**
> To further evaluate the efficacy of our approach, we employed the decoder-only Qwen2.5-3B-Instruct as our foundation model for conducting experiments. Specifically, we utilized our generated graph-based UniCoT as instruction input for Qwen2.5-3B-Instruct and incorporated our structural constrained adjacency matrix as additional prompts into the decoder-only model architecture. The model training still adhered to the next-token prediction paradigm via supervised fine-tuning training (SFT) with low-rank adaptation (LoRA). We implemented our method using llama-factory and provided corresponding configuration and code to facilitate the replication of our results. We are also adding new experimental configurations and codes to supplementary materials.
>
> As shown in the table below, our proposed UniCoTT method demonstrates consistent improvements across various tasks, including factual reasoning, multi-choice QA and mathematical reasoning (i.e., GSM8K) benchmarks, compared to the base model. The consistent improvements  demonstrate the effectiveness of our method on a wide range of different tasks. For more results and analysis, experimental details, and training loss curves, please refer to our revised manuscript.
>
> | Model | Factual Reasoning | Multi-choice QA | | | | Mathematical Reasoning |
> |-------|-------------------|-----------------|-----------------|-------|-------|---------------------|
> | | CREAK | CSQA2 | StrategyQA | CSQA | OBQA | GSM8K |
> | Base | 88.8 | 63.7 | 83.2 | 92.0 | 91.0 | 76.9 |
> | UniCoTT (Ours) | 91.5 | 75.4 | 88.7 | 95.0 | 92.9 | 79.2 |
>
> ---
>
> **[Q1]** Why focus on using an encoder as the student model?
>
> **[A2]**
> Small models with decoder-only architectures, such as those with 2B or 3B parameters, share similar architectures with larger decoder-only models. This architectural similarity enables more natural knowledge distillation methods between decoder-only models. For instance, knowledge can be distilled by aligning the logits output from large models with those from smaller models, or by reusing or concatenating parts of the large model's parameters to transfer capabilities to the smaller model.
> However, due to architectural differences, encoder-dependent small models cannot easily leverage these methods for knowledge transfer. Despite this limitation, encoder-dependent small models continue to be widely used in numerous practical scenarios, including discriminative tasks and resource-constrained edge devices.
>
> These considerations motivated us to design knowledge distillation strategies specifically for encoder-based models and classification tasks. Our approach aims to bridge the gap between the capabilities of large language models and the practical constraints of smaller, encoder-based models in real-world applications.

---

> ### Author Response · Authors · 2024-11-25
> **Thank you for your efforts.**
>
> Dear Reviewer AdyG,
>
> We are truly appreciative of the time and effort you have dedicated to reviewing our paper. Your thoughtful feedback and constructive suggestions are valuable to us. We have carefully addressed your comments in our rebuttal to enhance the quality of our work. As we approach the final days of the Author-Review Discussion period, we would like to ensure that all your concerns have been comprehensively addressed. Should there be any remaining questions or issues, we are willing to provide further clarification or additional revisions.Thank you once again for your insightful contributions to our work.
>
> Best regards,
>
> Team 5668

---

> > ### Comment · Reviewer_AdyG · 2024-11-27
> >
> > Thank you for your response, it has eased my concerns. I’ll keep my current score since it’s already a very positive score.

---

> > > ### Author Response · Authors · 2024-11-27
> > > **Thank You for Your Valuable Feedback on Our Manuscript**
> > >
> > > Dear Reviewer AdyG,
> > >
> > > We sincerely thank you for your valuable comments and insightful feedback on our work. We greatly appreciate the time and effort you have dedicated to reviewing our manuscript and for your positive evaluation of our work. Should you have any further questions or concerns, we are happy to respond promptly.
> > >
> > > Wishing you continued success in your career!
> > >
> > > Best regards,
> > >
> > > Paper Team #5668

---

### Meta-Review · Area_Chair_mFR7 · 2024-12-17

**Metareview:**

This paper proposes UniCoTT, a unified distillation framework to transfer the diverse reasoning structures with CoT to smaller language models such as BERT and RoBERTa. Firstly, UniCoT is proposed as a unified bridge of various CoT structures, which is constructed by iteratively prompting LLMs to produce explanations with correct answers. After that, a node-level supervised contrastive loss and a structural consistency loss are designed as part of the training objective. Experiments on multiple reasoning datasets verified the effectiveness of UniCoTT by surpassing the baseline methods by a large margin.

The proposed method is technically sound and intuitively makes sense. It is very interesting to transfer the knowledge from structured CoT texts into smaller models that can leverage rationale knowledge in a unified manner. Experimental results on several benchmarks show some improvement upon baselines and the authors conduct extensive ablation studies and analyses on various design choices.

On the other hand, there has been concern on the computational complexity of UniCoTT and its scalability, as well as its generalization on non-discriminative tasks. The latter is particularly important to incorporate given the topic of the work. Through the rebuttal phase, some of the other issues were addressed, while these aforementioned issues still remain.

**Additional Comments On Reviewer Discussion:**

There has been concern on the computational complexity of UniCoTT and its scalability, as well as its generalization on non-discriminative tasks. The latter is particularly important to incorporate given the topic of the work. Through the rebuttal phase, some of the other issues were addressed, while these aforementioned issues still remain.

---

### Decision · Program_Chairs · 2025-01-22

Accept (Poster)